# Switch-like and persistent memory formation in individual *Drosophila larvae*

**Amanda Lesar[1], Javan Tahir[1], Jason Wolk[1], Marc Gershow[1,2,3]***

[1]Department of Physics, New York University, New York, United States; [2]Center for Neural Science, New York University, New York, United States; [3]NYU Neuroscience Institute, New York University Langone Medical Center, New York, United States

**Abstract** Associative learning allows animals to use past experience to predict future events. The circuits underlying memory formation support immediate and sustained changes in function, often in response to a single example. Larval *Drosophila* is a genetic model for memory formation that can be accessed at molecular, synaptic, cellular, and circuit levels, often simultaneously, but existing behavioral assays for larval learning and memory do not address individual animals, and it has been difficult to form long-lasting memories, especially those requiring synaptic reorganization. We demonstrate a new assay for learning and memory capable of tracking the changing preferences of individual larvae. We use this assay to explore how activation of a pair of reward neurons changes the response to the innately aversive gas carbon dioxide ($CO_2$). We confirm that when coupled to $CO_2$ presentation in appropriate temporal sequence, optogenetic reward reduces avoidance of $CO_2$. We find that learning is switch-like: all-or-none and quantized in two states. Memories can be extinguished by repeated unrewarded exposure to $CO_2$ but are stabilized against extinction by repeated training or overnight consolidation. Finally, we demonstrate long-lasting protein synthesis dependent and independent memory formation.

## Introduction

Associative learning allows animals to use past experience to predict important future events, such as the appearance of food or predators, or changes in their environmental conditions (*Pavlov, 1927*; *Kandel et al., 2014*). The *Drosophila* larva is a favorable model system for the study of learning and memory formation (*Gerber et al., 2013*; *Widmann et al., 2018*; *Quinn and Dudai, 1976*; *Scherer et al., 2003*; *Apostolopoulou et al., 2013*; *Neuser et al., 2005*; *Saumweber et al., 2018*), with approximately 10,000 neurons in its representative insect brain. Widely available experimental tools allow manipulation of gene expression and introduction of foreign transgenes in labeled neurons throughout the *Drosophila* brain, including in the learning and memory centers (*Saumweber et al., 2018*; *Eichler et al., 2017*; *Li et al., 2014*; *Duffy, 2002*), whose synaptic connectivities can be reconstructed via electron microscopy (*Eichler et al., 2017*; *Eschbach et al., 2020a*; *Eschbach et al., 2020b*).

Larvae carry out complex behaviors including sensory-guided navigation (*Luo et al., 2010*; *Klein et al., 2015*; *Fishilevich et al., 2005*; *Asahina et al., 2009*; *Gomez-Marin and Louis, 2014*; *Gershow et al., 2012*; *Gomez-Marin et al., 2011*; *Sawin et al., 1994*; *Kane et al., 2013*; *Busto et al., 1999*; *Humberg et al., 2018*), which can be modified by learning (*Gerber et al., 2013*; *Scherer et al., 2003*; *Neuser et al., 2005*; *Widmann et al., 2018*). Larval *Drosophila* has long been a model for the study of memory formation, with a well-established paradigm developed to study associative memory formation through classical conditioning (*Gerber et al., 2013*; *Widmann et al., 2018*; *Schleyer et al., 2018*; *Scherer et al., 2003*; *Neuser et al., 2005*; *Gerber and Stocker, 2007*; *Apostolopoulou et al., 2013*; *Saumweber et al., 2018*; *Weiglein et al., 2019*). In this paradigm, larvae are trained and tested in groups, and learning is quantified by the difference in the olfactory

*For correspondence:
mhg4@nyu.edu

Competing interests: The authors declare that no competing interests exist.

**eLife digest** Brains learn from experience. They take events from the past, link them together, and use them to predict the future. This is true for fruit flies, *Drosophila melanogaster*, as well as for humans. One of the main questions in the field of neuroscience is, how does this kind of associative learning happen?

Fruit fly larvae can learn to associate a certain smell with a sugar reward. When a group of larvae learn to associate a smell with sugar, most but not all of them will approach that smell in the future. This shows associative learning in action, but it raises a big question. Did the larvae that failed to approach the smell fail to learn, or did they just happen to make a mistake finding the smell? Given another chance, would exactly the same larvae approach the smell as the first time? In other words, did all the larvae learn a little, or did some larvae learn completely and others learn nothing?

To find out, Lesar et al. built a computer-controlled maze to test whether individual fruit fly larvae liked or avoided a smell. Whenever a larva reached the middle of the Y-shaped maze, it could choose to go down one of two remaining corridors. One corridor contained air and the other carbon dioxide, a gas they would naturally avoid. Lesar et al. taught each larva to like carbon dioxide by activating reward neurons in its brain while filling the maze with carbon dioxide gas. Studying each larva as it navigated the maze revealed that they learn in a single jump, a 'lightbulb moment'. When Lesar et al. activated the reward neurons, the larva either 'got it' and stopped avoiding carbon dioxide altogether, or it did not. In the second case, it behaved as if it had received no training at all.

Classic and modern experiments on people suggest that humans might also learn in jumps, but research on our own brains is challenging. Fruit flies are an excellent model organism to study memory formation because they are easy to breed, and it is easy to manipulate their genetic code. Work in flies has already revealed many of the genes and cells responsible for learning and memory. But, to find the specific brain changes that explain learning, researchers need to know whether the animals they are examining have actually learned something. This new maze could help researchers to identify those individuals, making it easier to find out exactly how associative learning works.

preferences of differently trained groups of larvae. These assays quantify the effects of learning on a population level, but it is impossible to identify whether or to what extent an individual larva has learned.

New methods allow direct measurement of neural activity in behaving larvae (*Karagyozov et al., 2018*; *He et al., 2019*; *Vaadia et al., 2019*) and reconstruction of the connections between the neurons in a larva's brain (*Eichler et al., 2017*; *Eschbach et al., 2020a*; *Eschbach et al., 2020b*; *Takemura et al., 2017*; *Berck et al., 2016*), potentially allowing us to explore how learning changes the structure and function of this model nervous system. Using these tools requires us to identify larvae that have *definitively learned*. Recently, a device has been developed for assaying individual adult flies' innate (*Honegger et al., 2020*) and learned (*Smith et al., 2021*) olfactory preferences, but no comparable assay exists for the larval stage.

Further, to explore structural changes associated with learning, we need to form protein-synthesis dependent long-term memories (*Yin et al., 1995*; *Yin et al., 1994*; *Perazzona et al., 2004*). Larvae trained to associate odor with electric shock form memories that persist for at least 8 hr (*Khurana et al., 2009*). Odor-salt memories have been shown to partially persist for at least 5 hr (*Widmann et al., 2016*; *Eschment et al., 2020*) and can be protein-synthesis dependent (*Eschment et al., 2020*), depending on the initial feeding state of the larva. Overnight memory retention, whether or not requiring protein-synthesis, has not been demonstrated in the larva, nor has long-lasting retention of appetitive memories.

In this work, we demonstrate a new apparatus for in situ training and measurement of olfactory preferences for individual larvae. We use this assay to quantify appetitive memories formed by presentation of carbon dioxide ($CO_2$) combined with optogenetic activation of reward neurons. Using this device, we find that larvae are sensitive to both the timing and context of the reward presentation, that learning is quantized and all-or-none, and that repeated presentation of $CO_2$ without

reinforcer can erase a newly formed memory. We induce memories that persist overnight, and control whether these memories require protein synthesis through alteration of the training protocol.

## Results

### A Y-maze assay to characterize olfactory preferences of individual *Drosophila* larvae

Establishing the degree to which an individual larva seeks out or avoids an odorant requires repeated measurements of that larva's response to the odor. We developed a Y-maze assay (*Buchanan et al., 2015*; *Werkhoven et al., 2019*) to repeatedly test an individual's olfactory preference. The Y-mazes (*Figure 1A*) are constructed from agarose with channels slightly larger than the larvae, allowing free crawling only in a straight line (*Heckscher et al., 2012*; *Sun and Heckscher, 2016*). An individual larva travels down one channel and approaches the intersection with the other two branches of the maze. Here, the larva is presented with odorized air (or in this work, air containing $CO_2$) in one branch and pure air in the other. The larva then chooses and enters one of the two branches. This choice may be immediate or the result of a longer process in which the larva samples both channels and even reverses (*Figure 1—video 2*, *Figure 1—video 3*). When the larva reaches the end of its chosen channel, a circular chamber redirects it to return along the same channel to the intersection to make another choice. Custom computer vision software detects the motion of the larva while computer controlled valves manipulate the direction of airflow so that the larva is always presented with a fresh set of choices each time it approaches the intersection (*Figure 1A*, *Figure 1—video 1*).

We first sought to determine the suitability of this assay for measuring innate behavior. *Drosophila* larvae avoid carbon dioxide ($CO_2$) at all concentrations (*Faucher et al., 2006*; *Jones et al., 2007*; *Kwon et al., 2007*; *Gershow et al., 2012*). We presented larvae with a choice between humidified air and humidified air containing $CO_2$ each time they approached the central junction. At the 18% concentration used throughout this work, larvae with functional $CO_2$ receptors chose the $CO_2$-containing channel about 25% of the time. The probability of choosing the $CO_2$ containing channel increased as $CO_2$ concentration in that channel decreased (*Figure 1F*). *Gr63a*[1] (*Jones et al., 2007*) larvae lacking a functional $CO_2$ receptor were indifferent to the presence of $CO_2$ in the channel (*Figure 1B*), as were animals in which the $CO_2$ receptor neurons were silenced (Gr21a>Kir.21), indicating that larvae responded to the presence of $CO_2$ and not some other property of the $CO_2$ containing air stream. Silencing the Mushroom Body (OK107>Kir2.1) did not impair innate $CO_2$ avoidance.

### Pairing $CO_2$ presentation with optogenetic activation of a single pair of reward neurons eliminates $CO_2$ avoidance

Activation of the DAN-i1 pair of mushroom body input neurons has been shown to act as a reward for associative learning (*Saumweber et al., 2018*; *Thum and Gerber, 2019*; *Schleyer et al., 2020*; *Weiglein et al., 2019*; *Eschbach et al., 2020b*). In these experiments, the conditioned odor was innately attractive, but $CO_2$ is innately aversive. We wondered whether pairing DAN-i1 activation with $CO_2$ would lessen or even reverse the larva's innate avoidance of $CO_2$.

To train larvae in the same Y-maze used to measure preference, we manipulated the valves so that the entire chamber was either filled with humidified air or with humidified air mixed with additional $CO_2$, independent of the position of the larva, which was not tracked during training. At the same time, we activated DAN-i1 neurons expressing CsChrimson using red LEDs built in to the apparatus. For some larvae, we activated DAN-i1 when $CO_2$ was present (paired, *Figure 1D*). For others, we activated the reward neurons when only air was present (reverse-paired, *Figure 1D*). Each training cycle consisted of one 15 s $CO_2$ presentation and one 15 s air presentation, with DAN-i1 activated for the entirety of the $CO_2$ (paired) or air (reverse-paired) presentation phase. The training protocols schematized in *Figure 1D* were repeated for 20 successive cycles. Thus, for instance, in the reverse-paired scheme $CO_2$ offset at t=15s coincided with reward onset, and the reward offset at t=30s coincided with $CO_2$ onset at t=0 of the subsequent cycle.

For each larva, we first measured naive preference and then preference following training. We found that in the paired group, larvae became indifferent to $CO_2$ presentation following 20 training

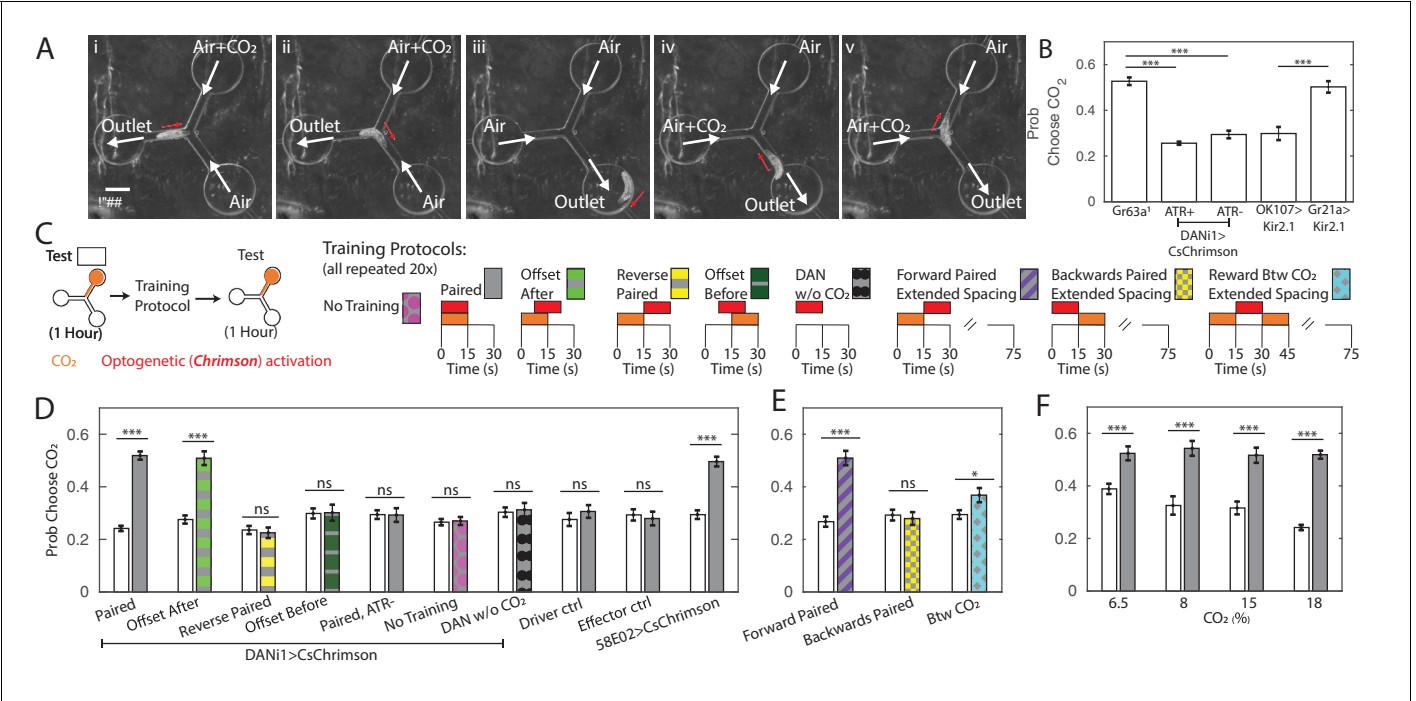

**Figure 1.** Y-maze assay to quantify innate and learned preference. (**A**) Image sequence of a larva making two consecutive decisions in the Y-maze assay. White arrows indicate direction of air flow; red arrow shows direction of larva's head. (**B**) Probability of choosing channel containing $CO_2$ without any training. (**C**) Schematic representation of experiments in (**D,E,F**). All larvae were tested in the Y-maze for 1 hr to determine initial preference and again following manipulation to determine a final preference. The manipulations were: Paired Training - reward in concert with $CO_2$ presentation, 15 s intervals, 20 repetitions; Offset After - reward presentation 7.5 s after $CO_2$ onset, 15 s intervals, 20 repetitions; Reverse-Paired Training - reward opposite $CO_2$ presentation, 15 s intervals, 20 repetitions; Offset Before - reward presentation 7.5 s before $CO_2$ onset, 15 s intervals, 20 repetitions; DAN Activation Without $CO_2$ - $CO_2$ is never presented, while reward is presented at 15 s intervals, 20 repetitions; no training - no manipulation between two testing periods; Forward Paired (extended spacing) - 15 s reward follows 15 s $CO_2$ presentation, followed by 60 s of air, 20 repetitions; Backwards Paired (extended spacing) - 15 s reward prior to 15 s $CO_2$ presentation, followed by 60 s of air, 20 repetitions; Reward Between $CO_2$ (extended spacing) - 15 s reward presentation between two 15 s $CO_2$ presentations, followed by 45 s of air, 20 repetitions. (**D**) Probability of choosing $CO_2$ containing channel before and after manipulation. All animals were fed ATR supplemented food, except those marked ATR-. (**E**) Probability of choosing $CO_2$ containing channel before and after training as a function of reward timing, in training protocols with extended air spacings. All animals were DANi1>CsChrimson and fed ATR. (**F**) Probability of choosing $CO_2$ containing channel before and after 20 cycles of paired training, as a function of $CO_2$ concentration, used both during training and testing. All animals were DANi1>CsChrimson and fed ATR. * $p<0.05$, ** $p<0.01$, *** $p<0.001$. The online version of this article includes the following video and source data for figure 1:

**Source data 1.** Spreadsheet containing each individual animal's decisions in temporal sequence.

**Figure 1—video 1.** Recording of a larva making 2 decisions within the Y-maze.

https://elifesciences.org/articles/70317#fig1video1

**Figure 1—video 2.** Recording of a larva before training, showing a sequence of decisions made at the Y-maze juncture.

https://elifesciences.org/articles/70317#fig1video2

**Figure 1—video 3.** Recording of a larva after training, showing a sequence of decisions made at the Y-maze juncture.

https://elifesciences.org/articles/70317#fig1video3

cycles (**Figure 1D**, DANi1>CsChrimson, Paired). We did not observe any change in preference in the reverse-paired group (DANi1>CsChrimson, Reverse-Paired). Nor did we observe a preference change following paired training for genetically identical animals not fed all-*trans*-retinal (ATR), a necessary co-factor for CsChrimson function (DANi1>CsChrimson, Paired ATR-). Animals fed ATR but not exposed to red light failed to show a preference shift (DANi1>CsChrimson, No Training). Larvae of the parent strains fed ATR and given paired training showed no preference shift (Effector Control, Driver Control). To control for possible effects of DAN-i1 activation, we activated DAN-i1 in 15 s intervals without presenting $CO_2$ at all during the training (DANi1>CsChrimson, DAN w/o $CO_2$); these larvae showed no shift in preference.

Taken together these results show that the change in $CO_2$ preference requires activation of the DAN-i1 neurons and is not due to habituation (*Twick et al., 2014*; *Das et al., 2011*; *Larkin et al., 2010*), red light presentation, or other aspects of the training protocol. In particular, the paired and reverse-paired group experienced identical $CO_2$ presentations and DAN-i1 activations with the only difference the relative timing between $CO_2$ presentation and DAN-i1 activation.

Activation of DAN-i1 coincident with $CO_2$ presentation decreased larvae's subsequent avoidance of $CO_2$. Formally, this admits two possibilities: the larva's preference for $CO_2$ increased because $CO_2$ was presented at the same time as the reward or because $CO_2$ predicted the reward. To test whether learning was contingent on coincidence or prediction, we carried out an additional set of experiments. As before, we first tested innate preference, then presented 20 alternating cycles of 15 s of $CO_2$ followed by 15 s of air. However, this time during the conditioning phase, we either activated DAN-i1 7.5 s *after* $CO_2$ onset, in which case $CO_2$ predicted DAN-i1 activation, or 7.5 s *before* $CO_2$ onset, in which case $CO_2$ predicted withdrawal of DAN-i1 activation.

In both cases, DAN-i1 was activated in the presence of $CO_2$ for 7.5 s and in the presence of air alone for 7.5 s. If learning depended only on the coincidence between reward and $CO_2$ presentations, both should be equally effective at generating a change in preference. In fact, we only found an increase in $CO_2$ preference following training in which the $CO_2$ predicted the reward (*Figure 1D*).

Next, we asked whether reward prediction alone was sufficient to establish a memory, or if coincidence between $CO_2$ and DAN-i1 activation was also required. We altered the training protocol to present 15 s of $CO_2$ followed by 60 s of air. Some larvae were rewarded by activation of DAN-i1 in the 15 s immediately following $CO_2$ presentation (Forward Paired), while others were rewarded in the 15 s immediately prior to $CO_2$ presentation (Backwards Paired). For a third group of larvae, $CO_2$ was presented both before and after reward presentation (reward between $CO_2$ presentations). At no time was DAN-i1 activated in the presence of $CO_2$, but in the first group $CO_2$ predicted DAN-i1 activation while in the others it did not. We found an increased $CO_2$ preference for animals in this first group only (*Figure 1E*), indicating that reward prediction is both necessary and sufficient for learning in this assay.

In other associative conditioning experiments using DAN-i1 activation as a reward, decreased attraction to the odor was observed in the reverse-paired groups (*Saumweber et al., 2018*; *Thum and Gerber, 2019*; *Schleyer et al., 2020*). In our experiments, we did not see any evidence of increased aversion in the reverse-paired groups.

Untrained larvae avoided $CO_2$. After 20 cycles of paired, offset-after, or forward-paired training, larvae no longer avoided $CO_2$, but they also did not seek it out. We wondered whether it might be possible to train larvae to develop an attraction to the innately aversive $CO_2$. In other contexts, reward via activation of 3 DANs (DAN-i1, DAN-h1, DAN-j1 - whether DAN-h1 is present in second instar larvae, used in this study, is presently unreported) labeled by the 58E02-Gal4 line has been reported to produce strong learning scores (*Saumweber et al., 2018*; *Rohwedder et al., 2016*; *Lyutova et al., 2019*; *Schleyer et al., 2020*). We repeated the training protocol, substituting 58E02 activation for DAN-i1 activation alone, but did not see an increased preference following training compared to DAN-i1 activation alone (*Figure 1D*, 58E02>CsChrimson).

Next, we asked how varying the $CO_2$ concentration might affect animals' performance in the assay. We presented lower concentrations of $CO_2$ both during the training and testing phases, and found that decreasing the $CO_2$ concentration decreased innate avoidance of $CO_2$. In all cases, following training, larvae lost avoidance to $CO_2$ but none showed statistically significant attraction (*Figure 1F*).

## Learning is quantized and all-or-none

We investigated how change in preference for $CO_2$ following associative conditioning with DAN-i1 activation depended on the amount of training. As in the previous experiments, we first measured the innate preference, then trained each larva using repeated cycles alternating pure and $CO_2$ containing air, while activating DAN-i1 in concert with $CO_2$ presentation. In these experiments, however, we varied the number of training cycles an individual larva experienced. We found that as a group, larvae that had experienced more training chose the $CO_2$ containing channel more often (*Figure 2A*).

Our data showed that increasing the amount of training increased overall preference for $CO_2$ up to a saturation point. But what was the mechanism for this change? Did each cycle of training

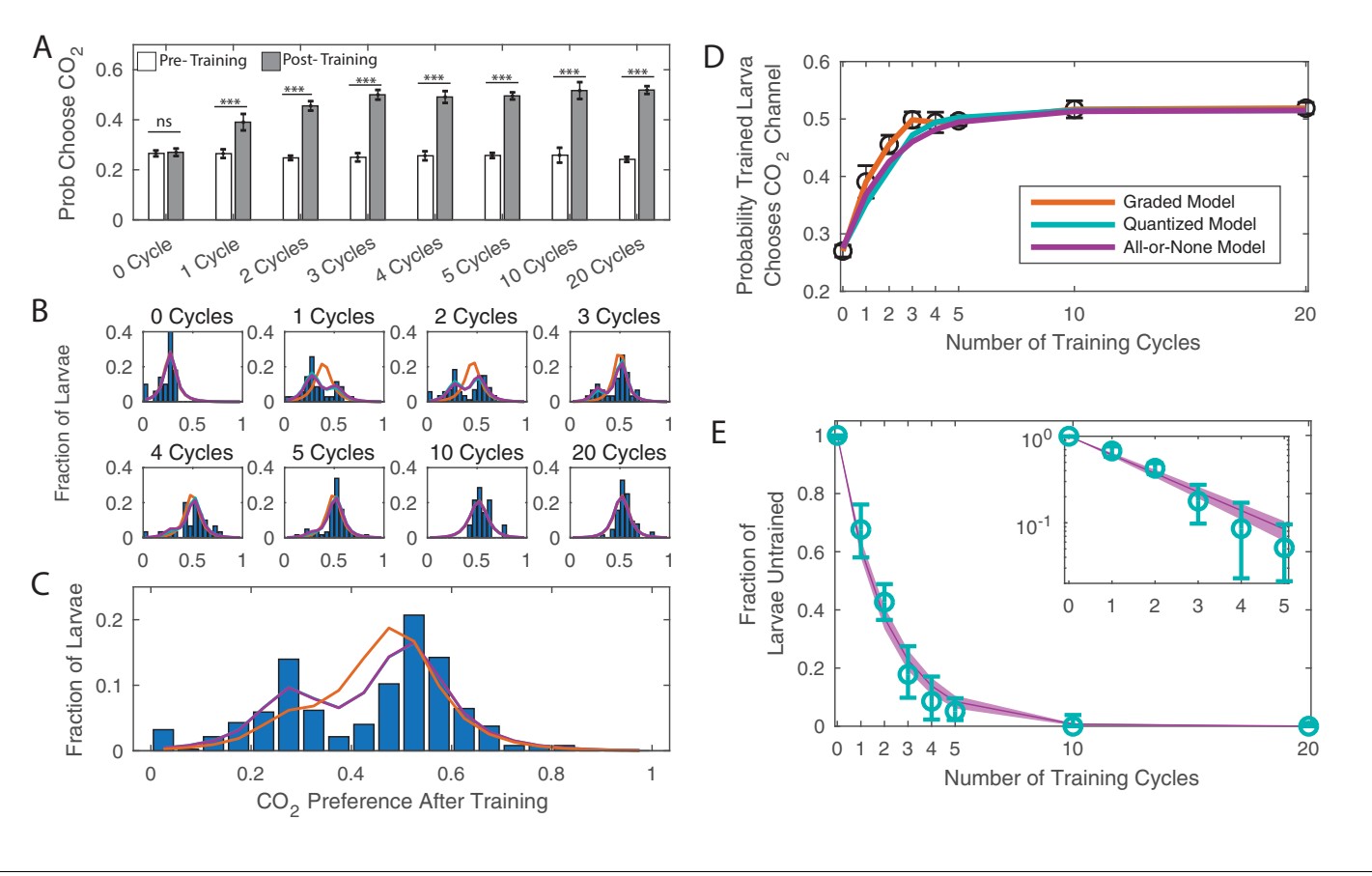

**Figure 2.** Dose dependence of learning DANi1>CsChrimson were given varying cycles of paired training (as in *Figure 1C*). (A) Probability of choosing $CO_2$ containing channel before and after training, as a function of amount of training. *** $p<0.001$. (B) Histograms of individual larva preferences after training, grouped by number of training cycles. (C) Histogram of individual larva preference after training for all larvae. (D) Population average probability of choosing $CO_2$ following training vs. dose. (E) Fraction of larvae untrained vs. number of training cycles. Teal: fit parameters and error ranges from quantized model, purple lines, prediction and error ranges from memoryless model. Note logarithmic y-axis on insert. (C–E) Orange: graded model prediction - post-training preference is represented by a single Gaussian distribution whose mean and variance depend on amount of training; Teal: quantized model prediction - post-training preference is represented by two fixed Gaussian distributions and the fraction of larvae in each population depends on the amount of training; Purple: all-or-none model prediction - post-training preference is represented by two fixed Gaussian distributions and the effect of a single training cycle is to train a fixed fraction of the remaining untrained larvae.

The online version of this article includes the following source data for figure 2:

**Source data 1.** Spreadsheet containing each individual animal's decisions in temporal sequence.

increase each larva's preference for $CO_2$ by some small amount, with the effect accumulating over repeated training (graded learning)? Or did some larvae experience a dramatic preference change – from naive to fully trained – with each cycle of training, with the number of fully trained larvae increasing with training repetitions (quantized learning)?

Either quantized or graded learning can explain the shift of mean preference of a population (*Gallistel et al., 2004*); to differentiate between the modes of learning, we examined repeated decisions made by individual animals, measurements that were impossible in previous larval assays. For each larva, we quantified the change in $CO_2$ preference before and after training. *Figure 2B* shows a histogram of larva preference (the fraction of times an individual larva chose the $CO_2$ containing channel) after training, grouped by the number of cycles of training a larva received.

Larvae that received no training (0 cycles) formed a single population that chose $CO_2$ 27% of the time. Larvae that were trained to saturation (20 training cycles) also formed a single group centered around 52% probability of choosing $CO_2$. Both the graded and quantized learning models make the same predictions for these endpoints, but their predictions vary starkly for the intermediate cases. A

graded learning model predicts that all larvae that received the same amount of training would form a single group whose mean preference for $CO_2$ would increase with increasing training. A quantized learning model predicts that larvae that have received the same amount of training will form two discrete groups ('trained' and 'untrained') with fixed centers whose means do not depend on the amount of training. With increased training an increasing fraction of larvae would be found in the trained group.

We fit the distributions of preference following conditioning to graded and quantized learning models. In the graded model, the preference was represented by a single Gaussian distribution whose mean and variance were a function of amount of training (orange, *Figure 2*). In the quantized model, the preference was represented by two Gaussian distributions; the fraction of larvae in each population was a function of the amount of training (teal, *Figure 2*).

We found that the data were better described by the quantized learning model (Table 4): larvae form two discrete groups, with the fraction in the trained group increasing with each cycle of additional training. The centers of the two groups do not vary with the amount of training, a point made most clear by considering the preference after training of all larvae taken together regardless of the amount of training received (*Figure 2C*), which shows two well defined and separated groups. From these data, we concluded that the effect of our associative conditioning on an individual larva is to either cause a discrete switch in preference or to leave the initial preference intact.

Next we asked what effect, if any, associative conditioning had on larvae that retained their innate preferences following training. Whether humans form associative memories gradually through repeated training or learn in an all-or-none manner has been the subject of debate in the Psychology literature (*Roediger and Arnold, 2012*); recent electrophysiological measurements in humans supports the all-or-none hypothesis (*Ison et al., 2015*). If learning is all-or-nothing, then if a larva has received training but has not yet expressed a behavioral switch, it is the same as if the larva has received no training at all. In this case, with every training cycle, regardless of past experience, every untrained larva will have the same probability of learning: $\rho$, and the effect of training can be described by a particularly simple equation

$$n_u(i+1) = n_u(i) - \rho n_u(i) \tag{1}$$

where $n_u(i)$ are the number untrained larvae following $i$ cycles of training. Note that $\rho$ can depend on the training protocol or other external variables, but it does not depend on the past training experiences of the larvae, and can be considered a fixed constant for a given experimental condition. The solution to this equation is an exponentially decaying population of untrained larvae. For a given initial population $n_u(0)$ of untrained larvae,

$$n_u(i) = (1-\rho)^i n_u(0) \tag{2}$$

Any so-called memoryless process like this produces an exponential decay of the initial population (*Apostol, 1969*). Meanwhile, processes with memory can produce other distributions. For example, if the training were *cumulative*, we would expect a threshold effect: as the number of cycles of training increased from 0, most larvae would initially remain untrained until a critical number of cycles ($n_c$) were reached and there would be a sudden shift to a mostly trained population. While a process with memory can also produce exponential decay (e.g. if each larva required a fixed $n_c$ cycles of training to learn, and $n_c$ was itself exponentially distributed), all memoryless processes must produce an exponential decay, and exponential decay is therefore an indicator of a memoryless (all-or-none) process.

Our fit to the quantized learning model produces an estimate of the fraction of larvae that remain untrained following training. We plotted the fraction of untrained larvae vs. number of training cycles and saw that the fraction of larvae in the untrained group exponentially decreased with increasing training (*Figure 2E*, note logarithmic y-axis on insert). We then fit the population distributions to an all-or-none quantized learning model in which the effect of a single training cycle was to train a fixed fraction of the remaining untrained larvae (purple, *Figure 2*). This model fit the data better than the graded learning model and almost as well as the original quantized learning model (in which the fraction of untrained larvae was fit separately to each group) despite having fewer parameters than either model. According to standard selection rules (BIC and AIC), the all-or-none quantized model best describes the data (Table 4).

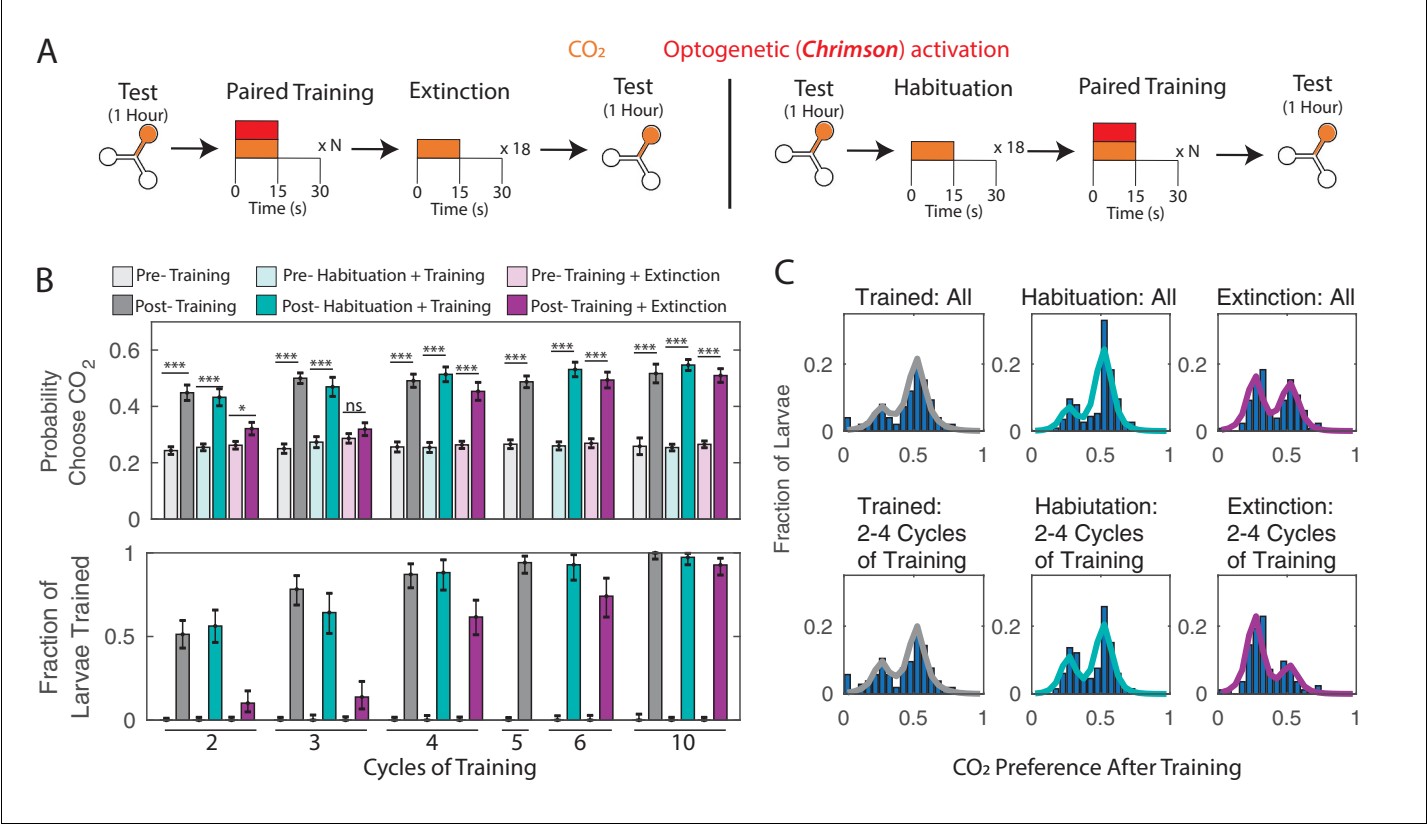

**Figure 3.** Memory extinction (**A**) Testing and training protocols for B,C. Training + Extinction: larvae were exposed to 18 cycles of alternating $CO_2$ and air following training. Habituation + Training: larvae were exposed to 18 cycles of alternating $CO_2$ and air prior to training. (**B**) Probability of choosing $CO_2$ containing channel (top) and fraction of larvae in trained group according to double Gaussian model fit (bottom) before and after training scheme. (**C**) Histograms of individual larva preference after training, for all larva and for larva trained with 2–4 training cycles. * $p<0.05$, ** $p<0.01$, *** $p<0.001$. The online version of this article includes the following source data and figure supplement(s) for figure 3:

**Source data 1.** Spreadsheet containing each individual animal's decisions in temporal sequence.
**Figure supplement 1.** After extinction, larvae can be trained again.
**Figure supplement 2.** Larvae population average response following training.
**Figure supplement 3.** Larvae given additional training between testing periods.

## Repeated exposure without reward following training leads to memory extinction

Reversal learning, in which the reward contingency is reversed, and extinction, in which the conditioned stimulus is presented without reward, experiments explore cognitive flexibility. Previous experiments with both adult *Drosophila* (*Tully et al., 1990*; *Ren et al., 2012*; *Wu et al., 2017*; *Vogt et al., 2015*) and larval (*Mancini et al., 2019*) *Drosophila* demonstrated a reversal learning paradigm. Extinction has been demonstrated in adult flies (*Felsenberg et al., 2017*; *Felsenberg et al., 2018*; *Schwaerzel et al., 2002*) but not in larvae.

To test for extinction, we again first measured an individual larva's $CO_2$ preference and then carried out associative conditioning for a given number (2-10) of training cycles. Next instead of immediately testing the larva's new preference for $CO_2$, we exposed the larva to an extinction phase – 18 cycles of alternating $CO_2$ and air without any optogenetic reward. Following the extinction period, we tested larvae as usual to measure their changed preference for $CO_2$. As a control against the effects of increased $CO_2$ exposure, we also performed habituation experiments, which were the same as the extinction experiments, except the 18 unrewarded cycles were presented *prior* to the rewarded training cycles. The extinction and habituation protocols are schematized in *Figure 3A*.

When we compared the 'habituated' groups of larvae to larvae trained for the same number of cycles without habituation or extinction, we found that unrewarded $CO_2$ presentation prior to

training had no effect on the eventual preference change (*Figure 3B*). This was unsurprising, as the initial testing period already offered a number of unrewarded $CO_2$ presentations. In contrast, unrewarded $CO_2$ presentations *following* training reversed the effect of training; for small (2 or 3 cycles) amounts of training, the reversal was almost complete (*Figure 3B*).

We previously observed that associative conditioning produced a discrete and quantized change in $CO_2$ preference. Here, we found that extinction following training greatly reversed the effects of conditioning. We wondered whether larvae that had been subject to both training and extinction reverted to their original $CO_2$ preference or to an intermediate state. In the former case, we would expect to see a bimodal distribution of preference change following training and extinction, while in the latter we would see a third group of larvae. This group would be most evident in experiments where two to four cycles of training were followed by extinction, as these had the largest deficit in the fraction of trained larvae compared to habituated larvae that received the same amount of training. We examined the preferences of all larvae following two to four cycles of training, grouped by whether they were normally trained, habituated, or subject to extinction (*Figure 3C*). In all cases, we observed two groups with the same central means and no evidence of a third intermediate group. We concluded that larvae subject to training then extinction reverted to their 'untrained' state.

We wondered whether larvae would still learn if they received additional training directly following extinction. As before, we measured the innate preference, presented three paired training cycles followed by the extinction phase. At this point, based on our previous results, larvae would have returned to their initial innate avoidance of $CO_2$. We then immediately presented three more paired training cycles before behavioral testing (*Figure 3—figure supplement 1A*). We found that following this training-extinction-training protocol, both the population preference for $CO_2$ (*Figure 3—figure supplement 1B*) and the fraction of larvae trained (*Figure 3—figure supplement 1C*) were comparable to larvae that had been trained three times without extinction cycles.

Given the relatively short duration of training and the ability of unrewarded $CO_2$ presentations to extinguish prior training, we wondered whether larvae might change their $CO_2$ preferences over the course of the hour-long post-training behavioral readout. In particular, might the apparent threshold of 50% attraction be an artifact due to a short period of attraction to $CO_2$ followed by a longer period of indifference or modest avoidance?

To test for a short period of increased attraction immediately following training, we reanalyzed the results of experiments with 2, 5, and 20 cycles of paired training. In each case, we compared the initial 10 min of the post-training choice assay to the final 50 min (*Figure 3—figure supplement 2A*) and found no significant difference between the initial and final periods for any of these training conditions. We then compared the mean preference over the first five choices (representing five unrewarded $CO_2$ presentations) made by each larva to the mean preference in the remainder of the experiment and again found no significant difference (*Figure 3—figure supplement 2B*). Breaking the behavioral readout into equal 15 min periods also reveals no strong temporal signal (*Figure 3—figure supplement 2C–E*).

Finally, we developed a new protocol to minimize the possible effects of extinction over the course of the behavioral readout, using the fact that training following extinction can re-establish a lost memory (*Figure 3—figure supplement 1*). We trained each larva with 10 paired cycles (5 min of training), then tested their preference for 15 min, then presented another 10 paired training cycles followed by another 15 min of testing, for a total of 4 training and testing blocks (*Figure 3—figure supplement 3*). The results were comparable to when we presented a single training block followed by an hour-long test period. Thus, we concluded that the apparent limit of 50% population preference to $CO_2$ following training was not due to the long time-scale of the behavioral readout.

## Larvae can retain memory overnight; the type of memory formed depends on the training protocol

Studies in adult (*Tully et al., 1990*; *Yin et al., 1995*; *Margulies et al., 2005*) and larval (*Honjo and Furukubo-Tokunaga, 2005*; *Honjo and Furukubo-Tokunaga, 2009*; *Widmann et al., 2016*; *Khurana et al., 2009*; *Eschment et al., 2020*; *Aceves-Piña and Quinn, 1979*) *Drosophila* have identified distinct memory phases: short-term memory (STM), middle-term memory (MTM), long-term memory (LTM), and anesthesia-resistant memory (ARM). LTM and ARM are consolidated forms of memory controlled by partially separate molecular and anatomical pathways (*Isabel et al., 2004*; *Jacob and Waddell, 2020*; *Wu et al., 2012*). ARM is resistant to anesthetic agents (*Quinn et al.,*

*1974*); LTM requires cAMP response element-binding protein (CREB)-dependent transcription and de-novo protein synthesis, while ARM does not (*Yin et al., 1995*; *Perazzona et al., 2004*). Adults have been shown to retain memories for up to a week (*Yin et al., 1995*). Larvae trained to associate odor with electric shock form memories that persist for at least 8 hr (*Khurana et al., 2009*). Odor-salt memories have been shown to persist for at least 5 hr (*Widmann et al., 2016*; *Eschment et al., 2020*) and can be either ARM or LTM, depending on the initial feeding state of the larva.

We sought to determine whether we could create consolidated memories that would persist overnight, and if so, whether these memories represented ARM or LTM. As in previously described experiments, we first tested each larva's individual preference in the Y-maze assay, trained it to associate $CO_2$ presentation with DAN-i1 activation, and then measured its individual preference again following training. After this second round of testing, we removed the larva from the apparatus and placed it on food (without ATR) overnight. The next day, we placed the larva back in the Y-maze and again tested its preference for $CO_2$, without any additional training.

We found that following 20 cycles of training, larvae became indifferent to $CO_2$ and this indifference persisted to the next day. Similarly, we found that most larvae switched preference following five cycles of training and retained that preference overnight. Larvae that received no training or 20 cycles of unpaired training had no change in $CO_2$ preference immediately following training or the next day (*Figure 4B*).

We had previously shown two cycles of training caused roughly half the larva to change preference immediately after training. We decided to use this partition to verify a correlation between immediate and long-term memories; we expected that larvae initially in the 'trained' group would also form a 'trained' group the following day. However, while we found that two cycles of training were sufficient to cause some larvae to become indifferent to $CO_2$ immediately following training, when we tested these larvae the next day, we found that all had reverted to their initial avoidance of $CO_2$.

There were two possible explanations for this reversion. Perhaps, two cycles of training were sufficient to form a short term memory, but more training was required to induce a long-term memory. Or perhaps the *testing* period, in which larvae were exposed repeatedly to $CO_2$ without reward, reversed the two-cycle training. To control for the latter, we modified the experimental protocol. We tested each larva's innate preference, presented two training cycles, and then immediately removed the larva to food overnight, without any further testing. When we tested these larvae the next day, we found that they showed decreased avoidance of $CO_2$. This indicated that two cycles of training were sufficient to form a memory lasting overnight, but that immediate exposure to unrewarded $CO_2$ following this short training interval likely reversed the effects of training, an effect we observed in *Figure 3*. When larvae were trained for 20 cycles, omitting the testing had no effect on these larvae's preferences the following day.

To confirm that extinction could explain the failure to form a persistent memory, we exposed larvae to three cycles of paired training, then 18 cycles of extinction (as in *Figure 3*) and then removed them to food overnight before testing their preferences the next day. As expected, these larvae avoided $CO_2$ as much the next day as they did prior to training (*Figure 4B*, Ext Post-Train).

We wondered whether memories that had consolidated overnight would be more resistant to extinction. We repeated the previous experiment with a single modification. As before, we tested the larva's initial preference and trained it with three cycles of rewarded $CO_2$ presentation. This time, we immediately removed the larva to food following training. The next day, we returned the larva to the Y-maze and presented the extinction phase of 18 unrewarded $CO_2$ presentations prior to testing for $CO_2$ preference. We found that in this case, larvae still expressed an increased preference for $CO_2$ despite the extinction phase (*Figure 4B*, Ext Pre-Test). The only difference between the two experiments was whether we attempted extinction immediately after training or the next day. Thus, we concluded that overnight consolidation made memories more resistant to extinction.

ARM can be distinguished from LTM because the latter requires de novo protein synthesis and can be disrupted by ingestion of the translation-inhibitor cycloheximide (CXM). To incorporate CXM feeding, we modified our protocols. Instead of raising larvae on ATR supplemented food, we raised them on standard food and then fed them with ATR supplemented yeast paste for 4 hr prior to the experiment (ATR+/CXM-). For some larvae (ATR+/CXM+), we also added CXM to the yeast paste. In this way, we could be sure that if ATR+/CXM+ larvae ingested enough ATR to allow for CsChrimson activation of DAN-i1, they must have also ingested CXM as well. To further verify CXM

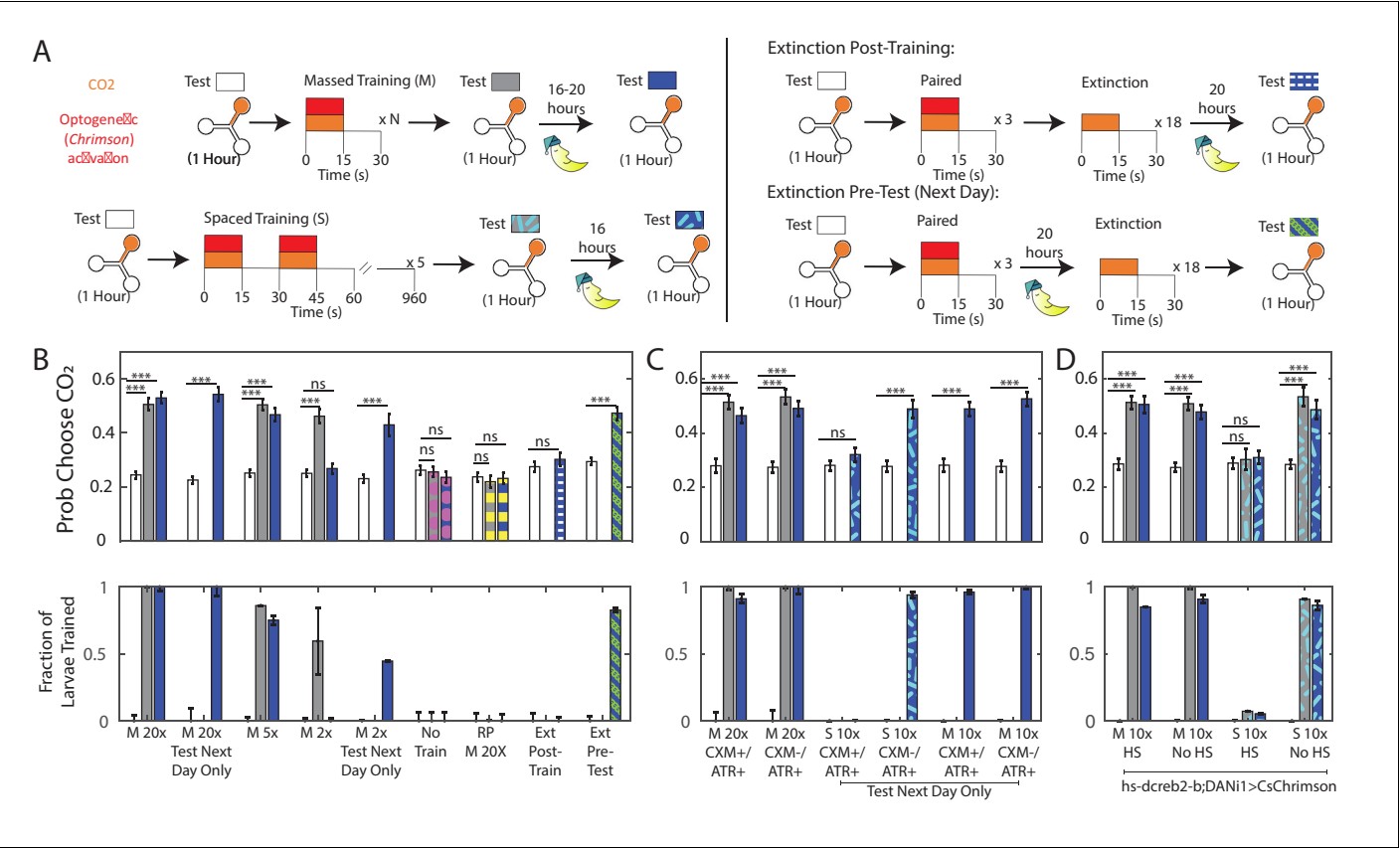

**Figure 4.** Memory retention overnight. (**A**) Testing and training protocols. Except where indicated, larvae were tested, trained immediately after testing, tested again, then placed on food overnight and tested the following day. For extinction experiments, larvae were trained three times, and then exposed to 18 cycles of alternating $CO_2$ and air either immediately following training or prior to testing the next day. (**B,C,D**) Probability of choosing $CO_2$ containing channel (top) and fraction of larvae in trained group according to double Gaussian model fit (bottom) prior to training, immediately following training, and the next day. When the center bar is missing, larvae were not tested immediately following training but instead removed immediately to food. M Nx = massed training, N repetitions, S 10x = spaced training 10 total pairings, RP = reverse paired (see *Figure 1C*), No Train = no training. Larvae in (**B,C**) were DANi1>CsChrimson. Larvae in (**D**) were DANi1>hs-dCREB2-b;CsChrimson. Larvae were raised on food containing ATR, except for ATR+/CXM-, ATR+/CXM+ larvae who were fed ATR supplemented yeast paste (without/with cycloheximide) for 4 hr prior to initial testing. For reverse-paired (RP) and no training schemes, see *Figure 1B*. * p<0.05, ** p<0.01, *** p<0.001.

The online version of this article includes the following source data for figure 4:

**Source data 1.** Spreadsheet containing each individual animal's decisions in temporal sequence.

ingestion, we placed ATR+/CXM+ and ATR+/CXM- larvae on clean food and allowed them to continue development. 95% of ATR+/CXM- larvae pupated, while only 45% of ATR+/CXM+ larvae pupated.

Following the 4 hr feeding period, ATR+/CXM+ and ATR+/CXM- larvae were treated identically. As in the previously described experiments, we first tested each larva's individual preference in the Y-maze assay, trained the larva 20 times to associate $CO_2$ presentation with DAN-i1 activation, and then measured its individual preference again following training. After this second round of testing, we removed the larva from the apparatus and placed it on food (without ATR or CXM) overnight. The next day we placed the larva back in the Y-maze and again tested its preference for $CO_2$, without any additional training.

We found that performances tested immediately and 16 hr after training were both unaffected by CXM treatment. Following 20 cycles of training, larvae from both groups (ATR+/CXM+; ATR+/CXM-) became indifferent to $CO_2$ and this indifference persisted to the next day (*Figure 4C*). This suggests that the memory formation was independent of de novo protein synthesis.

In adult *Drosophila*, whether ARM or LTM is formed depends on the training protocol (*Tully et al., 1990*; *Tully et al., 1994*; *Yin et al., 1995*; *Yu et al., 2006*; *Bouzaiane et al., 2015*).

'Massed' training, in which all conditioning occurs in rapid sequence without rest intervals, results in ARM, while 'spaced' training, in which the conditioning occurs in blocks separated by intervals of time, produces LTM. Our training protocol more closely resembles massed training, so it seemed sensible that it would produce ARM. To see if we could instead develop LTM, we established a spaced training protocol. Larvae received two paired cycles of training, followed by a 15-min interval of air-presentation only; this sequence was repeated five times. To keep the total length of the experiment within a (covid-related) limited daily time window, we did not test the larvae immediately after training but only the next day.

Prior to spaced training, both ATR+/CXM- and ATR+/CXM+ larvae avoided $CO_2$ to the same degree. We found that 1 day following spaced training, ATR+/CXM+ larvae continued to avoid $CO_2$, while ATR+/CXM- larvae did not. This indicated that spaced training formed a memory whose retention was disrupted by CXM. To verify that spacing the trials was essential to forming a protein-synthesis dependent memory, we duplicated the experiments exactly, except we presented 10 cycles of training *en masse*, rather than spacing them. In this case, both ATR+/CXM- and ATR+/CXM- larvae expressed learned indifference to $CO_2$ 1 day following training (*Figure 4C*).

As an alternate to CXM feeding, LTM (but not ARM) formation can also be disrupted through use of hs-dCREB2-b, a heat-shock inducible dominant-negative repressor of transcription mediated by dCREB2-a. (*Perazzona et al., 2004*; *Yin et al., 1995*). Specifically, in adult flies expressing hs-dCREB2-b, memory retention is disrupted in a heat-shock-dependent manner following spaced, but not massed training (*Yin et al., 1995*). We therefore repeated our long-term memory experiments in larvae that in addition to expressing Chrimson in DAN-i1 neurons also carried the hs-dCREB2-b transgene. Massed and spaced training were carried out as previously described, using larvae raised on ATR supplemented food, except that some larvae (HS) received a 30 min heat-shock (at 37 C), followed by a 30 min recovery period (at 25 C) immediately prior to the beginning of the experiment (i.e. prior to the initial testing of naive preference). Preference for $CO_2$ was tested prior to training, immediately following training, and the next day, following an overnight rest on food without ATR.

We found that, congruent with our CXM experiments, the day after spaced training, heat-shocked larvae (*Figure 4D,S* 10x, HS) avoided $CO_2$ to the same extent they did prior to training, while larvae that were not heat-shocked (*Figure 4D,S* 10X, No HS) retained learned indifference; larvae that received massed training (*Figure 4D,M* 10x, HS and No HS) retained their learned indifference overnight, regardless of heat-shock. These results are consistent with similar experiments in adult flies (*Yin et al., 1994*; *Yu et al., 2006*).

Immediately following spaced training (80 min after the initiation of the spaced training protocol), heat-shocked larvae continued to avoid $CO_2$, showing that memory formation was impaired on a relatively short timescale. This is consistent with previous work in the larva, where dCREB2-b expression induced memory deficits beginning 30 min following a single 30 min training cycle (*Honjo and Furukubo-Tokunaga, 2005*), and immediately following 125 min of spaced training (*Widmann et al., 2016*). In those experiments, neither training protocol was shown to induce a persistent long-term memory.

## Discussion

In this work, we demonstrated a new apparatus for training individual larvae and assessing their olfactory preferences. Compared to the existing paradigm, our assay allows for measuring individual animals' changes in preference due to training, allows for greater control of the temporal relation between the conditioned and unconditioned stimuli, and does not require any handling of the animals between training and testing.

In our assay, larvae learned in a switch-like (all-or-none two-state quantized) manner. The learning process was better described as a sudden transition between states than as a graded change in preference, and each cycle of training (presentation of $CO_2$ coupled with reward) either caused a state transition or did not. Pigeons, rats, and rabbits have all been shown to experience sudden performance increases in learning tasks, suggesting quantized learning may be a generalized phenomenon (*Gallistel et al., 2004*). We found no evidence of a cumulative effect of prior training in the probability that a given cycle of training would induce a state transition in larvae that had not already transitioned. We did, however, find evidence that repeated cycles of training stabilized memories against

later extinction effected by presentation of $CO_2$ without reward. These measurements were enabled by our assay's ability to track individual preferences over the course of the entire experiment.

We directly tested the ability of unrewarded $CO_2$ presentations to extinguish a just-formed memory by presenting $CO_2$ without air immediately following training (*Figure 3*). We also indirectly measured the effects of extinction due to unrewarded $CO_2$ presentation during the hour-long behavioral test (*Figure 3—figure supplement 2*). Following two cycles of training, immediate presentation of 18 unrewarded $CO_2$ cycles abolished the formed memory (*Figure 3B*), but without this direct extinction protocol, we saw no evidence of extinction over the course of the hour-long behavioral test (*Figure 3—figure supplement 2A–C*). It is perhaps unsurprising that rapid and consistent unrewarded presentations immediately following training are more effective at extinguishing a memory than the later and more varied unrewarded presentations during the behavioral test. But following two cycles of training, the behavioral test *does* prevent expression of the formed memory the next day (*Figure 4B*). This could show that the unrewarded presentations during behavioral testing are too late and/or sporadic to prevent immediate memory expression but do prevent the transition to more long-lived ARM. Further study will be required to confirm this. Our apparatus can precisely control the timing and nature of both rewarded and unrewarded presentations to probe different phases of memory formation and consolidation.

We found that larvae trained in our assay retained memories overnight: 16–20 hr. When training was presented all at once, these memories were not disrupted by ingestion of the protein-synthesis inhibitor cycloheximide or induction of the transcrption repressor dCREB2-b, while when training was spaced over time, cycloheximide feeding and dCREB2-b induction both prevented long duration memory formation. Thus, we identified spaced training as producing long-term memory (LTM) and the massed training as producing anesthesia-resistant memory (ARM). These results are the first demonstrations that larvae can retain memories overnight; they are entirely congruent with observations in adult flies.

We explored how the order of $CO_2$ and reward presentations affected learning. We found that for larvae to learn, $CO_2$ onset must occur coincident with or before reward onset, but that it was neither necessary nor sufficient for $CO_2$ and reward to be presented together at the same time. While we assume that the same neural mechanism underlies learning in the 'paired', 'offset after' (*Figure 1D*) and 'forward-paired' (*Figure 1E*) paradigms, it is at least formally possible that the mechanism might be different in these contexts. Most of the work in this paper used the 'paired' protocol; it would be interesting to test in the future whether the 'forward-paired' protocol produces memories that differ in their resistance to extinction or in their long-term persistence.

Our results using the 'reverse paired' (*Figure 1D*) and 'backwards paired' (*Figure 1E*) protocols differed from previous reports. In other assays, presenting the reward (including via activation of DAN-i1) prior to presenting the conditioned odor results in *decreased* attraction/increased avoidance (*Schleyer et al., 2020*; *Saumweber et al., 2018*) of that odor. We found that such 'reverse-pairings' neither increased nor decreased a larva's avoidance of $CO_2$. There are a number of differences, most significantly our new behavioral assay and our use of the innately aversive $CO_2$ as the conditioned stimulus that might account for the discrepancy.

While this work does not directly speak to the neural mechanism behind the change in preference, it is congruent with the evolving model of learning in *Drosophila*. In this model, different Mushroom Body Output Neurons (MBONs) promote approach or avoidance (*Aso et al., 2014*; *Eschbach and Zlatic, 2020*; *Perisse et al., 2013*; *Owald et al., 2015*; *Owald and Waddell, 2015*; *Hige et al., 2015*) and synapse onto a convergence neuron that integrates their activities (*Eschbach and Zlatic, 2020*). Prior to learning, aversive and appetitive MBONs are thought to receive similar drives from Kenyon Cells (KCs) that respond to specific olfactory signals. That is, in response to a stimulus, the activities of MBONs representing these opposite valences are initially balanced, and behavior is governed by an innate preference to that odor, controlled by neuronal circuits external to the MB (*Aso et al., 2014*; *Eschbach and Zlatic, 2020*). Aversive and appetitive learning depress the odor drive to appetitive and aversive MBONs, respectively: learning that a stimulus is appetitive weakens the connection between KCs encoding that stimulus and the avoidance MBONs, promoting approach, while aversive conditioning weakens the connection between KCs and approach MBONs, promoting avoidance (*Aso et al., 2014*; *Owald and Waddell, 2015*; *Eschbach and Zlatic, 2020*).

According to this model, presentation of $CO_2$ coincident with or prior to the activation of DAN-i1 reduces the ability of $CO_2$ to excite one or more aversive MBONs, likely including MBON-i1, which encodes avoidance (*Eschbach and Zlatic, 2020*) and is postsynaptic to DAN-i1 (*Eichler et al., 2017*). This results in an appetitive drive from the MB that cancels out the innate avoidance pathway. Why in our experiments the learned appetitive drive appears to exactly cancel but not overcome the innate aversion should be the subject of further study; it may be a simple coincidence or artifact of the experimental protocol, or it may reflect more profound circuit principles.

Understanding memory formation at the circuit and synaptic levels simultaneously is a heroic task, even aided by the larva's numerically simple nervous system and the tools (including EM-reconstruction) available in the larva. The work here represents progress toward this goal. We demonstrate long-term protein synthesis dependent memory, implying that memories are encoded in synaptic change. Our assay allows us to precisely identify those individuals who have formed long-term memories. Animals are found in only two behavioral states: innate avoidance or learned indifference; this likely reflects two discrete states of the underlying neural circuit.

Our associative conditioning paradigm pairing $CO_2$ presentation with DAN-i1 activation has experimental advantages for circuit-cracking. The conditioned stimulus is sensed by a single pair of genetically identified sensory neurons; the unconditioned stimulus is provided by activation of a single pair of genetically identified reward neurons whose connectivity has been fully reconstructed (*Schleyer et al., 2020*). How the larva navigates in response to $CO_2$ presentation has been described in detail (*Faucher et al., 2006*; *Gershow et al., 2012*; *Gepner et al., 2015*; *Gepner et al., 2018*), as has how neurons downstream of DAN-i1 and the KCs contribute to navigational decision making (*Eichler et al., 2017*; *Saumweber et al., 2018*; *Thum and Gerber, 2019*; *Schleyer et al., 2020*). This is a particularly favorable starting point to understand how synaptic plasticity due to associative conditioning leads to changes in circuit function that effect changed behavioral outcomes.

## Conclusion

We introduced a Y-maze assay capable of measuring the olfactory preferences of individual larval *Drosophila* and of in situ associative conditioning. We found that when larvae learn to associate $CO_2$ with reward neuron activation, the result is a switch from innate avoidance to learned indifference, with no intervening states. We demonstrated a protocol to form stable protein-synthesis dependent long term memories. This provides a strong starting point for 'cracking' a complete olfactory learning circuit.

# Materials and methods

**Key resources table**

| Reagent type (species) or resource | Designation | Source or reference | Identifiers | Additional information |
|---|---|---|---|---|
| Genetic reagent (*D. melanogaster*) | w[1118]; P{y[+t7.7]w[+mC]=20XUAS-IVS-CsChrimson.mVenus}attP2 (w;;UAS-CsChrimson) | Bloomington Stock Center | RRID:BDSC_55136 | |
| Genetic reagent (*D. melanogaster*) | SS00864 split-Gal4 (DAN-i1-Gal4) | *Saumweber et al., 2018* | | Gift of Marta Zlatic, Janelia Research Campus |
| Genetic reagent (*D. melanogaster*) | w[*]; Gr63a[1] | Bloomington Stock Center | RRID:BDSC_9941 | |
| Genetic reagent (*D. melanogaster*) | w[1118]; P{y[+t7.7]w[+mC]=GMR58E02-GAL4}attP2 (GMR58E02-Gal4) | Bloomington Stock Center | RRID:BDSC_41347 | |
| Genetic reagent (*D. melanogaster*) | w;hs-dCREB2-b 17–2 | *Yin et al., 1995* | FlyBase_ FBti0038019 | Gift of Jerry Chi-Ping Yin, University of Wisconsin, Madison |

*Continued on next page*

*Continued*

| Reagent type (species) or resource | Designation | Source or reference | Identifiers | Additional information |
|---|---|---|---|---|
| Genetic reagent (*D. melanogaster*) | w[*]; P{w[+mW.hs]=GawB} ey[OK107]/In(4)ci[D], ci[D] pan[ciD] sv[spa-pol] (OK107-Gal4) | Bloomington Stock Center | RRID:BDSC_854 | |
| Genetic reagent (*D. melanogaster*) | w[*]; P{w[+mC]=UAS-Hsap\KCNJ2.EGFP}7 (UAS-kir2.1) | Bloomington Stock Center | RRID:BDSC_6595 | |
| Genetic reagent (*D. melanogaster*) | w[*]; P{w[+mC] =Gr21a-GAL4.C} 133t52.1 (Gr21a-Gal4) | Bloomington Stock Center | RRID:BDSC_23890 | |
| Software, algorithm | livetracker | github.com/GershowLab/ TrainingChamber (copy archived at URL swh:1:rev: e2a7ccc4e8d845e 6cac59d3b2f344cca826c4727, *Lesar, 2021*) | This work | |

## Crosses and genotypes

### Larva collection

Flies of the appropriate genotypes (*Table 1*) were placed in 60 mm embryo-collection cages (59–100, Genessee Scientific) and allowed to lay eggs for 6 hr at 25C on enriched food media (Nutri-Fly German Food, Genesee Scientific). For all experiments except otherwise specified, the food was supplemented with 0.1 mM all-trans-retinal (ATR, Sigma Aldrich R2500). Cages were kept in the dark during egg laying. When eggs were not being collected for experiments, flies were kept on plain food at 18C.

Petri dishes containing eggs and larvae were kept at 25C in the dark for 48–60 hr. Second instar larvae were separated from food using 30% sucrose solution and washed in water. Larvae were selected for size. Preparations for experiments were carried out in a dark room.

### Y-maze

We used SLA three-dimensional printing to create microfluidic masters for casting (*Karagyozov et al., 2018*; *Chan et al., 2015*). Masters were designed in Autodesk Inventor and printed on an Ember three-dimensonal printer (Autodesk) using black prototyping resin (Colorado Photopolymer Solutions). After printing, masters were washed in isopropyl alcohol, air-dried, and baked at 65C for 45 min to remove volatile additives and non-crosslinked resin. 4% agarose (Apex Quick Dissolve LE Agarose, Cat #20-102QD, Genesee Scientific) was poured over the masters and

**Table 1.** Crosses used to generate larvae for experiments throughout this work.
For strain information, see key resource table.

| Figure | Designation | Female parent | Male parent |
|---|---|---|---|
| 1 | Gr63a[1] | | w;Gr63a[1] |
| 1 | OK107>Kir2.1 | UAS-Kir2.1-GFP | OK107-Gal4 |
| 1 | Gr21a>Kir2.1 | UAS-Kir2.1-GFP | Gr21a-Gal4 |
| 1-4 | DANi1>CsChrimson | w;;UAS-CsChrimson | SS00864 |
| 1 | Driver ctrl | SS00864 | |
| 1 | Effector ctrl | w;;UAS-CsChrimson | |
| 1 | 58E02>CsChrimson | w;;UAS-CsChrimson | 58E02-Gal4 |
| 4 | hs-dcreb2-b;DANi1>CsChrimson | w;hs-dcreb2-b;UAS-CsChrimson | SS00864 |

allowed to solidify; then mazes were removed from the mold. Agarose Y-mazes were stored in tap water before use.

The mazes are 1 mm in depth. Each channel is 1.818 mm in length and 0.4 mm in width, and ends in a circular chamber (radius = 1 mm) which redirects larva back to the intersection. An inlet channel (depth = 0.1 mm, length = 1.524 mm, width = 0.1 mm) to the circular chamber connects to tubing for our network of air, $CO_2$, and vacuum sources.

## Behavioral experiments

Individual larvae were selected for size and placed into a Y-maze using a paintbrush. The Y-maze was placed into a PDMS (Sylgard 184, 10:1 base:curing agent) base, where tubing was secured. The Y-maze and base were encased in a dark custom-built box. Larvae were monitored under 850 nm infrared illumination (Everlight Electronics Co Ltd, HIR11-21C/L11/TR8) using a Raspberry Pi NoIR camera (Adafruit, 3100), connected to a Raspberry Pi microcomputer (Raspberry Pi 3 Model B+, Adafruit, 3775). Experiments were recorded using the same camera, operating at 20 fps. Eight copies of the assay were built, to assay the behaviors of multiple larvae in parallel.

Pressure for air, $CO_2$, and vacuum were controlled at the sources (for vacuum regulation: 41585K43, McMaster-Carr; for pressure regulation: 43275K16, McMaster-Carr). $CO_2$ and air were humidified through a bubble humidifier. Vacuum, air, and $CO_2$ tubing to individual assays were separated through a block manifold after pressure control and humidification (BHH2-12, Clippard).

The $CO_2$ concentration was controlled by a resistive network of tubing connected to the air and $CO_2$ sources. This inexpensive alternative to a mass-flow controller produced a stable ratio of $CO_2$ to air that was consistent from day to day and independent of the overall flow rate. The direction of flow was controlled by solenoid pinch valves (NPV2-1C–03–12, Clippard), actuated by a custom circuit we designed.

Custom computer vision software detected the location of the larva in real time. Based on the larva's location, computer controlled valves manipulated the direction of airflow so that the larva was always presented with a fresh set of choices each time it approached the intersection. The software randomly decided which channel would contain air and which contained air mixed with $CO_2$.

In each maze, one channel was selected to be the outlet for flow and the other two were inlets. An individual larva began in the outlet channel and approached the intersection of the Y-maze, then chose to enter either an inlet branch containing air with $CO_2$ or an inlet branch containing air only. When the larva's full body entered the chosen channel, software recorded the larva's choice of channel. When the larva reached the end of that channel and entered the circular chamber, valves switched to turn off $CO_2$ and to switch vacuum to the channel containing larva, making that channel the outlet. The $CO_2$ remained off (the larva experienced only pure air) until the larva exited the circular chamber. When the larva exited the circular chamber and proceeded towards the intersection, $CO_2$ was introduced to one randomly selected inlet channel.

Software recorded the location of the larva at every frame (approx 20 Hz); the direction of airflow in the maze (which channel(s) had air; which channel had $CO_2$ mixed with air, if any; and which channel had vacuum); and all decisions the larva made. We recorded when larvae entered or left a channel, and whether that channel presented $CO_2$. Larvae could take three actions as they approached the intersection: choose the channel containing air with $CO_2$ (scored as APPROACH); choose the channel containing pure air (scored as AVOID); or move backwards into their original channel before they reach the intersection. If a larva backed up and reentered the circular chamber it departed from before reaching the intersection, the software reset and presented the larva with a fresh set of choices when it next left the circle. We did not score backing up as a choice of either $CO_2$ or air.

Following an hour of testing, larvae were trained in the same Y-maze assay used to measure preference. During the training period, unless described otherwise, each 30 s training cycles alternated 15 s of $CO_2$ presentation, where both inlet channels contained a humidified mix of $CO_2$ and air; followed by 15 s of air presentation, where both inlet channels had humidified air alone. This cycle was repeated some number of times (specified for each experiment in the figures). Red LEDs (Sun LD, XZM2ACR55W-3) integrated into the setup were used to activate CsChrimson synchronously with $CO_2$ presentation (paired) or air presentation (reverse-paired).

The volume of the flow chamber was 11.68 $mm^3$ and the volume of the tubing downstream of the valves is approximately 214 $mm^3$. The flow rate exceeded 560 $mm^3/s$, and the state of the chamber was taken to be the same as the state of valves.

Following training, larvae were tested for one hour in an identical scheme to that previously described for the naive measurement.

After larvae were placed into the Y-maze, larva were left in the maze in the dark for a minimum of 5 min. If a larva was not moving through the maze after 5 min, the larva was replaced before the experiment began. If larvae stopped moving through the maze during the first hour of testing, larvae were removed from the maze before training and results were discarded. This happened infrequently (approximately 5% of experiments).

## Protocol for timing dependence experiments

For experiments in *Figure 1D*, reward presentation was offset from $CO_2$ onset. 30 s training cycles alternated 15 s of $CO_2$ presentation, where both channels contained a mix of $CO_2$ and air; followed by 15 s of air presentation, where neither channel had $CO_2$. Red LEDs are used to activate CsChrimson for 15 s. For some larvae, reward onset occurred 7.5 s after $CO_2$ presentation; for others, reward onset occurred 7.5 s before $CO_2$ presentation. For all experiments of this type, larvae were presented with 20 cycles of training.

For experiments in *Figure 1E*, 75-second training cycles alternated 15 s of $CO_2$ presentation, where both inlet channels contained a mix of $CO_2$ and air with 60 s of air presentation. For some larvae, reward presentation occurred immediately following $CO_2$ termination for 15 s. For others, reward presentation occurred 15 s prior to $CO_2$ onset, and reward presentation was terminated upon $CO_2$ presentation. For a third group of larvae, we rewarded larvae for 15 s between two $CO_2$ presentations. In this case, 15 s of $CO_2$ presentation was followed by 15 s of reward presentation in the absence of $CO_2$, followed by another 15 s of $CO_2$ presentation. After the second presentation, there was a 30 s air gap before the cycle repeated. For all experiments of these types, larvae were presented with 20 cycles of training.

## Habituation and extinction protocols

For experiments in *Figure 3*, we used either an extinction or habituation protocol during training. For both types, larvae were tested for 1 hr to determine their innate $CO_2$ preference in the method described above.

For extinction experiments, larvae were trained in the same Y-maze used to measure preference. 30 s training cycles alternate 15 s of $CO_2$ presentation, where both channels contain a mix of $CO_2$ and air; followed by 15 s of air presentation, where neither channel had $CO_2$. Red LEDs were used to activate CsChrimson synchronously with $CO_2$ presentation. This training cycle was repeated some number of times (specified for each experiment above). Immediately after training, we presented the larva with 18 cycles of repeated $CO_2$/air exposure (15 s of $CO_2$ followed by 15 s of air; repeat) with no reward pairing. After these extinction cycles, larva preference for $CO_2$ was tested for one hour.

Habituation experiments were done exactly as for extinction experiments, except that the 18 unrewarded cycles of repeated $CO_2$/air exposure were presented immediately prior to the training cycles.

For experiments in *Figure 4B*, we tested each larva's initial preference for one hour, then presented three rewarded paired training cycles. For some larvae ('Extinction Post-Train'), we then immediately presented 18 extinction cycles, removed the larvae to food overnight as described above, and then tested their preferences for one hour the next day. For another set of larvae ('Extinction Pre-Test'), we removed the larvae to food immediately following training. The next day, after the larvae were cleaned and inserted into the Y-maze, they were exposed to 18 extinction cycles immediately prior to testing their $CO_2$ preferences for 1 hr.

## Overnight memory formation

For the memory retention experiments of *Figure 4*, testing and training followed identical procedures as above to establish larva preference. After the second round of testing testing, the larvae were removed from the Y-maze assay with a paintbrush and transferred to an individual 4% agar plate (30 mm, FB0875711YZ Fisher Scientific), with yeast paste added. Larvae were kept in the dark at 18 C for approximately 20 hr. Prior to experiments the next day, larva were removed from the agar plate and washed in water before being placed in a new Y-maze. Larvae were then tested for

$CO_2$ preference for one hour as previously described. In all experiments in which larvae were removed from the apparatus and later retested, they were placed in the same apparatus but with a new agar Y-maze. Out of 443 larvae placed on agar plates to be tested the following day, 439 larvae were recovered and retested. The four lost larvae were excluded from analysis.

## Cycloheximide feeding protocol

For specified experiments in section *Figure 4*, larva were raised on ATR- food plates at 25C for 48 hr. Second instar larvae were separated from food using 30% sucrose solution and washed in water. Four hours prior to experiments, larvae were transferred to an agar dish with yeast paste for feeding. Yeast paste was made with either a solution of 35 mM cycloheximide (CXM, Sigma Aldrich C7698) and 0.1 mM all-trans-retinal (ATR, Sigma Aldrich R2500) in 5% sucrose (ATR+/CXM+); or 0.1 mM ATR in 5% sucrose (ATR+/CXM-). To verify CXM ingestion, we placed ATR+/CXM+ and ATR+/CXM- larvae not selected for experiments back on clean food and allowed them to continue development. 95% of ATR+/CXM- larvae pupated, while only 45% of ATR+/CXM+ larvae pupated. Before the experiment, larvae were transferred to an empty petri dish and washed with tap water before being placed into a maze. Except where noted, the same experimental protocol was followed as for non-CXM overnight memory.

## Protocol for cycloheximide experiments

For the CXM experiments in section *Figure 4*, larvae were trained with either a massed or spaced training protocol. The 20x massed training protocol was as previously described for other experiments in *Figure 4*.

In the 10x spaced training protocol, larvae were first tested for 1 hr to determine their initial $CO_2$ preference. They then received two cycles of paired DAN-i1 activation with $CO_2$ presentation (15 s of $CO_2$ presentation paired with reward, followed by 15 s of air presentation), followed by 15 min of air presentation. This was then repeated five times (10 activations total). In these experiments, we did not test the larvae immediately following training but instead removed them to food and tested their preferences the next day only.

The 10x massed training protocol was identical to the 10x spaced training protocol, except training consisted of 10 sequential cycles of paired DAN-i1 activation with $CO_2$ presentation (15 s of $CO_2$ presentation paired with reward, followed by 15 s of air presentation, repeated 10 times). As in the spaced training experiments, larvae were removed to food immediately following training, and their preferences were tested the next day only.

## hs-dCREB2-b heat-shock protocol

Petri dishes with larvae were placed in an oven at 37°C for 30 min. The petri dish was placed in a water bath in the oven and covered to preserve humidity and to ensure ATR+ larvae were kept in the dark. The larvae were then removed to 25°C for a 30 min recovery period before experiments. Experiments began immediately after the recovery period. Larvae were kept in the dark at all times during this protocol.

Larvae were tested for $CO_2$ preference prior to training, immediately following training, and the next day, after being kept overnight on food without ATR. Some larvae in the spaced training groups were not active immediately following training. In the heat-shocked group, 11 out of 24 larvae made less than five decisions during the immediate test; in the not-heat-shocked group, 6 out of 22 larvae made less than five decisions during the immediate test. For this scheme, larvae were in the Y-maze for longer than previous experiments, as the spaced training protocol is 80 min (compared to approximately 10 min or less for the standard massed training). All larvae were retested the following day, even if the larva did not make many decisions when tested immediately following training. After the overnight rest period, all non-heat-shocked and 20/24 heat-shocked larvae were active in the final test period. Inactive larvae were included in the analysis and in the bootstrapping of error bars, but contributed little to the population measure because of the few decisions made.

## Development of initial protocols

There are a number of parameters that can be adjusted in our assay, including the identity and concentration of gas used as a CS, the concentration and timing of ATR feeding, the period, duty cycle, and number of CS and US presentations, and duration of behavioral readouts before and after training. We began with our normal protocol for optogenetic activation (*Gepner et al., 2015*; *Gepner et al., 2018*): eggs were laid on ATR supplemented food, and larvae were raised in the dark. We somewhat arbitrarily chose a 30 s, 50% duty cycle applied for 20 cycles as our standard for paired training presentation; we began with DANi1>CsChrimson based on previous work (*Saumweber et al., 2018*; *Thum and Gerber, 2019*; *Schleyer et al., 2020*; *Weiglein et al., 2019*; *Eschbach et al., 2020b*), and the fact that CsChrimson can be activated via red light without provoking a strong visual response. We then adjusted the $CO_2$ concentration to maximize the contrast between $CO_2$ preference before and after training. From this basic platform, we changed as little as possible while manipulating the parameter of interest, for example we maintained the 30 s 50% duty cycle paired training while changing the number of cycles, or we maintained 20 cycles while varying the temporal sequence of CS and US presentation, or we used exactly the same 30 s, 50% duty cycle, 20 cycle paired protocol while changing the driver to RF58E02.

## Data analysis

The probability of choosing the $CO_2$ containing channel was scored for individual larvae and for populations as

$$\text{Prob choose CO}_2 = \frac{\#\text{APPROACH}}{\#\text{APPROACH} + \#\text{AVOID}} \tag{3}$$

The population average was determined by dividing the total number of times any larva in the population chose the $CO_2$ containing channel by the total number of times any larva chose either channel. In other words, larvae that made more decisions contributed more heavily to the average.

The number of larva and total number of approach and avoid decisions made by larvae for each type of experiment is shown in *Table 2*. Error bars for 'probability choose $CO_2$' data displays and all significance tests in the figures were generated by bootstrapping.

For each experimental set, we performed the bootstrapping as follows. If there were X larvae from that experiment, we selected X larvae with replacement from that set. Then, from each larvae selected, we selected with replacement from the decisions that larvae had made. For example, if the larvae had made Y 'approach' and Z 'avoid' decisions, we selected (Y+Z) decisions with replacement from that set to represent the larvae. We then calculated the population average from this generated set of animals. We generated 10,0000 numerical replicates using this bootstrapping method. Error bars were the standard deviation of these replicates. Note that in each replicate, the same animals were included in each (e.g. trained and untrained) group.

A p-value $p<x$ indicates that at least $x$ fraction of these replicates ended with the same ranking result (e.g. $p<0.01$ between trained and untrained would indicate that in at least 9900 out of 10,000 replicates, the trained group had a larger $CO_2$ preference than the untrained group or vice versa). These p-values are included in the 'Hierarchical Bootstrap' column of *Table 3*.

We also performed a non-hierarchical bootstrap, in which animals were resampled but their decisions were not, preserving any correlations between decisions. In this case, we generated 10,000 numerical replicates by selecting with replacement from that set of larvae; the actual sequence of decisions made by the resampled larvae was then used without further resampling. A p-value $p<x$ indicates that at least $x$ fraction of these replicates ended with the same ranking result. These p-values are included in the 'Bootstrap Animal Only' column of *Table 3*. In *Table 3*, we also show p-values for the Fisher's exact test, which treats every decision as independent, and the Mann-Whitney u-test, which treats every larva in each group as a discrete measurement and does not account for differing numbers of decisions made by larvae.

To fit the data in *Figure 2* to various models, we used a maximum-likelihood approach. First we grouped the larvae according to the number of cycles ($n_c$) of training they received. In each group, for each larva we quantified the number of decisions made following training. The number of decisions made by the $j^{th}$ larva that received $n_c$ cycles of training was $n(n_c,j)$ and the fraction of times the

**Table 2.** Data for experiments in *Figure 1*, *Figure 2*, *Figure 3*, and *Figure 4*.

# Larva: number of individual larvae tested for experiment type; # Approach Pre-Train: total number of times all larvae chose the channel containing air with $CO_2$ prior to training; # Avoid Pre-Train: total number of times all larvae chose the channel containing pure air prior to training; # Approach Post-Train: total number of times all larvae chose the channel containing air with $CO_2$ after the indicated training scheme; # Avoid Post-Train: total number of times all larvae chose the channel containing pure air after the indicated training scheme; # Approach Next Day: total number of times all larvae chose the channel containing air with $CO_2$ during testing approximately 20 hr after training; # Avoid Next: total number of times all larvae chose the channel containing pure air during testing approximately 20 hr after training. All tests were 1 hr (for each larva).

| Experiment | Genotype | # Larva | # Approach Pre-Train | # Avoid Pre-Train | # Approach Post-Train | # Avoid Post-Train | # Approach Next Day | # Avoid Next Day |
|---|---|---|---|---|---|---|---|---|
| *Figure 1B* | | | | | | | | |
| Gr63a[1] | Gr63a[1] | 44 | 831 | 745 | - | - | - | - |
| DANi1> CsChrimson, ATR+ | DANi1> CsChrimson | 159 | 1714 | 4978 | - | - | - | - |
| DANi1> CsChrimson, ATR- | DANi1> CsChrimson | 16 | 256 | 614 | - | - | - | - |
| *Figure 1D* | | | | | | | | |
| Paired | DANi1> CsChrimson | 64 | 561 | 1760 | 936 | 868 | - | - |
| Offset After | DANi1> CsChrimson | 20 | 288 | 757 | 316 | 305 | - | - |
| Reverse Paired | DANi1> CsChrimson | 29 | 315 | 1022 | 154 | 530 | - | - |
| Offset Before | DANi1> CsChrimson | 19 | 218 | 512 | 136 | 315 | - | - |
| Paired, ATR- | DANi1> CsChrimson | 16 | 256 | 614 | 127 | 307 | - | - |
| No Training | DANi1> CsChrimson | 50 | 578 | 1599 | 479 | 1295 | - | - |
| DAN w/o $CO_2$ | DANi1> CsChrimson | 16 | 260 | 597 | 161 | 354 | - | - |
| Driver ctrl | SS00864 | 17 | 110 | 289 | 158 | 358 | - | - |
| Effector ctrl | UAS-CsChrimson | 18 | 214 | 516 | 114 | 294 | - | - |
| 58E02> CsChrimson | 58E02> CsChrimson | 21 | 380 | 912 | 493 | 501 | - | - |
| *Figure 1E* | DANi1> CsChrimson | | | | | | | |
| Forward Paired | | 22 | 181 | 496 | 350 | 337 | - | - |
| Backwards Paired | | 18 | 181 | 438 | 124 | 320 | - | - |
| Btw $CO_2$ | | 23 | 272 | 652 | 165 | 283 | - | - |
| *Figure 1F* | DANi1> CsChrimson | | | | | | | |
| 6.5% | | 19 | 361 | 568 | 319 | 290 | - | - |
| 8% | | 27 | 256 | 567 | 295 | 255 | - | - |
| 15% | | 19 | 170 | 368 | 249 | 233 | - | - |
| 18% | | 64 | 561 | 1760 | 936 | 868 | - | - |
| *Figure 2A* | DANi1> CsChrimson | | | | | | | |
| 0 Cycles | | 50 | 578 | 1599 | 479 | 1295 | - | - |
| 1 Cycles | | 35 | 218 | 606 | 317 | 495 | - | - |
| 2 Cycles | | 87 | 840 | 2552 | 1081 | 1292 | - | - |
| 3 Cycles | | 31 | 310 | 930 | 686 | 686 | - | - |
| 4 Cycles | | 32 | 245 | 712 | 493 | 511 | - | - |
| 5 Cycles | | 63 | 863 | 2491 | 975 | 993 | - | - |
| 10 Cycles | | 14 | 100 | 287 | 154 | 144 | - | - |
| 20 Cycles | | 64 | 561 | 1760 | 936 | 868 | - | - |
| *Figure 3B* | DANi1> CsChrimson | | | | | | | |
| 2 Cycles, Training | | 87 | 840 | 2552 | 1081 | 1292 | - | - |
| 2 Cycles, Habituation + Training | | 30 | 385 | 1127 | 422 | 554 | - | - |

*Table 2 continued on next page*

*Table 2 continued*

| Experiment | Genotype | # Larva | # Approach Pre-Train | # Avoid Pre-Train | # Approach Post-Train | # Avoid Post-Train | # Approach Next Day | # Avoid Next Day |
|---|---|---|---|---|---|---|---|---|
| 2 Cycles, Training + Extinction | | 30 | 336 | 946 | 375 | 793 | - | - |
| 3 Cycles, Training | | 30 | 308 | 924 | 675 | 679 | - | - |
| 3 Cycles, Habituation + Training | | 18 | 222 | 591 | 260 | 294 | - | - |
| 3 Cycles, Training + Extinction | | 26 | 279 | 695 | 195 | 416 | - | - |
| 4 Cycles, Training | | 30 | 225 | 659 | 490 | 502 | - | - |
| 4 Cycles, Habituation + Training | | 18 | 239 | 701 | 372 | 352 | - | - |
| 4 Cycles, Training + Extinction | | 27 | 384 | 1074 | 394 | 475 | - | - |
| 5 Cycles, Training | | 63 | 863 | 2491 | 975 | 993 | - | - |
| 6 Cycles, Habituation + Training | | 19 | 266 | 758 | 367 | 324 | - | - |
| 6 Cycles, Training + Extinction | | 18 | 253 | 687 | 309 | 317 | - | - |
| 10 Cycles, Training | | 14 | 100 | 287 | 154 | 144 | - | - |
| 10 Cycles, Habituation + Training | | 30 | 406 | 1193 | 607 | 503 | - | - |
| 10 Cycles, Training + Extinction | | 30 | 426 | 1180 | 401 | 386 | - | - |
| *Figure 4B* | DANi1> CsChrimson | | | | | | | |
| 20x | | 28 | 380 | 1172 | 509 | 499 | 459 | 409 |
| 20x (Only Test Next Day) | | 14 | 224 | 768 | - | - | 296 | 250 |
| 5x | | 29 | 472 | 1427 | 488 | 480 | 404 | 461 |
| 2x | | 42 | 514 | 1537 | 594 | 693 | 201 | 548 |
| 2x (Only Test Next Day) | | 22 | 209 | 696 | - | - | 213 | 283 |
| No Train | | 20 | 316 | 889 | 187 | 544 | 104 | 337 |
| RP 20x | | 21 | 282 | 905 | 121 | 430 | 109 | 361 |
| Ext Post-Train | | 23 | 181 | 477 | - | - | 158 | 365 |
| Ext Pre-Test | | 31 | 417 | 1002 | - | - | 385 | 429 |
| *Figure 4C* | DANi1> CsChrimson | | | | | | | |
| M 20x (CXM+/ATR+) | | 20 | 110 | 282 | 252 | 237 | 237 | 272 |
| M 20x (CXM-/ATR+) | | 17 | 159 | 419 | 271 | 236 | 228 | 235 |
| S 20x (CXM+/ATR+) | | 23 | 191 | 486 | - | - | 150 | 316 |
| S 20x (CXM-/ATR+) | | 20 | 197 | 511 | - | - | 254 | 264 |
| M 10x (CXM+/ATR+) | | 23 | 136 | 345 | - | - | 331 | 344 |
| M 10x (CXM-/ATR+) | | 20 | 175 | 454 | - | - | 419 | 375 |
| *Figure 4D* | DANi1> hs-dCREB2-b; CsChrimson | | | | | | | |
| M 10x HS | | 21 | 175 | 434 | 392 | 370 | 253 | 246 |
| M 10x No HS | | 22 | 248 | 656 | 367 | 353 | 451 | 490 |
| S 10x HS | | 24 | 172 | 420 | 68 | 156 | 153 | 339 |
| S 10x No HS | | 22 | 294 | 736 | 212 | 184 | 335 | 352 |

**Table 3.** p-Values for experiments in *Figure 1*, *Figure 2*, *Figure 3*, and *Figure 4*.

P-values for experiments were calculated: Bootstrap - p-values calculated as explained in Materials and methods; Fisher - p-values calculated using Fisher's exact test; U-test - p-values calculated using two-sided Mann–Whitney U test. Unless otherwise noted, p-values are calculated between pre-train and post-train data. A shaded row indicates not all tests reach the same significance level (out of ns, $p < 0.05$, $p < 0.01$, $p < 0.001$).

| Experiment | Genotype | Hierarchical Bootstrap | Bootstrap Animal Only | Fisher | U-test |
|---|---|---|---|---|---|
| *Figure 1B* | | | | | |
| Gr63a[1]/DANi1> CsChrimson, ATR+ | | $<10^{-4}$ | $<10^{-4}$ | $<10^{-4}$ | $<10^{-4}$ |
| Gr63a[1]/DANi1> CsChrimson, ATR- | | $<10^{-4}$ | $<10^{-4}$ | $<10^{-4}$ | $<10^{-4}$ |
| *Figure 1D* | | | | | |
| Paired | DANi1> CsChrimson | $<10^{-4}$ | $<10^{-4}$ | $<10^{-4}$ | $<10^{-4}$ |
| Offset After | DANi1> CsChrimson | $<10^{-4}$ | $<10^{-4}$ | $<10^{-4}$ | $<10^{-4}$ |
| Reverse Paired | DANi1> CsChrimson | 0.3429 | 0.2689 | 0.6166 | 0.9379 |
| Offset Before | DANi1> CsChrimson | 0.4479 | 0.4373 | 0.9479 | 0.9770 |
| Paired, ATR- | DANi1> CsChrimson | 0.4762 | 0.4315 | 1.000 | 0.2658 |
| No Training | DANi1> CsChrimson | 0.4066 | 0.3664 | 0.7726 | 0.9835 |
| DAN w/o $CO_2$ | DANi1> CsChrimson | 0.3935 | 0.3102 | 0.7173 | 0.4852 |
| Driver ctrl | SS00864 | 0.3106 | 0.0313 | 0.3411 | 0.3977 |
| Effector ctrl | UAS-CsChrimson | 0.3383 | 0.2361 | 0.6336 | 0.8366 |
| 58E02> CsChrimson | 58E02> CsChrimson | $<10^{-4}$ | $<10^{-4}$ | $<10^{-4}$ | $<10^{-4}$ |
| *Figure 1C* | DANi1> CsChrimson | | | | |
| Forward Paired | | $<10^{-4}$ | $<10^{-4}$ | $<10^{-4}$ | $<10^{-4}$ |
| Backwards Paired | | 0,3368 | 0.163 | 0.6801 | 0.1939 |
| Btw $CO_2$ | | 0.0107 | 0.0001 | 0.006543 | 0.0003257 |
| *Figure 1D* | DANi1> CsChrimson | | | | |
| 6.5% | | $<10^{-4}$ | $<10^{-4}$ | $<10^{-4}$ | $<10^{-4}$ |
| 8% | | $<10^{-4}$ | $<10^{-4}$ | $<10^{-4}$ | $<10^{-4}$ |
| 15% | | $<10^{-4}$ | $<10^{-4}$ | $<10^{-4}$ | $<10^{-4}$ |
| 18% | | $<10^{-4}$ | $<10^{-4}$ | $<10^{-4}$ | $<10^{-4}$ |
| *Figure 2A* | DANi1> CsChrimson | | | | |
| 0 Cycles | | 0.4132 | 0.3647 | 0.7726 | 0.9835 |
| 1 Cycles | | 0.0003 | $<10^{-4}$ | $<10^{-4}$ | 0.0591 |
| 2 Cycles | | $<10^{-4}$ | $<10^{-4}$ | $<10^{-4}$ | $<10^{-4}$ |
| 3 Cycles | | $<10^{-4}$ | $<10^{-4}$ | $<10^{-4}$ | $<10^{-4}$ |
| 4 Cycles | | $<10^{-4}$ | $<10^{-4}$ | $<10^{-4}$ | $<10^{-4}$ |
| 5 Cycles | | $<10^{-4}$ | $<10^{-4}$ | $<10^{-4}$ | $<10^{-4}$ |
| 10 Cycles | | $<10^{-4}$ | $<10^{-4}$ | $<10^{-4}$ | $<10^{-4}$ |
| 20 Cycles | | $<10^{-4}$ | $<10^{-4}$ | $<10^{-4}$ | $<10^{-4}$ |
| *Figure 3B* | DANi1> CsChrimson | | | | |
| 2 Cycles, Training | | $<10^{-4}$ | $<10^{-4}$ | $<10^{-4}$ | $<10^{-4}$ |
| 2 Cycles, Habituation + Training | | $<10^{-4}$ | $<10^{-4}$ | $<10^{-4}$ | $<10^{-4}$ |
| 2 Cycles, Training + Extinction | | 0.0117 | 0.0020 | 0.001339 | 0.04743 |
| 3 Cycles, Training | | $<10^{-4}$ | $<10^{-4}$ | $<10^{-4}$ | $<10^{-4}$ |
| 3 Cycles, Habituation + Training | | $<10^{-4}$ | $<10^{-4}$ | 0.0007459 | $<10^{-4}$ |
| 3 Cycles, Training + Extinction | | 0.1133 | 0.0176 | 0.1763 | 0.03069 |
| 4 Cycles, Training | | $<10^{-4}$ | $<10^{-4}$ | $<10^{-4}$ | $<10^{-4}$ |

*Table 3 continued on next page*

*Table 3 continued*

| Experiment | Genotype | Hierarchical Bootstrap | Bootstrap Animal Only | Fisher | U-test |
|---|---|---|---|---|---|
| 4 Cycles, Habituation + Training | | $<10^{-4}$ | $<10^{-4}$ | $<10^{-4}$ | $<10^{-4}$ |
| 4 Cycles, Training + Extinction | | $<10^{-4}$ | $<10^{-4}$ | $<10^{-4}$ | $<10^{-4}$ |
| 5 Cycles, Training | | $<10^{-4}$ | $<10^{-4}$ | $<10^{-4}$ | $<10^{-4}$ |
| 6 Cycles, Habituation + Training | | $<10^{-4}$ | $<10^{-4}$ | $<10^{-4}$ | $<10^{-4}$ |
| 6 Cycles, Training + Extinction | | $<10^{-4}$ | $<10^{-4}$ | $<10^{-4}$ | $<10^{-4}$ |
| 10 Cycles, Training | | $<10^{-4}$ | $<10^{-4}$ | $<10^{-4}$ | $<10^{-4}$ |
| 10 Cycles, Habituation + Training | | $<10^{-4}$ | $<10^{-4}$ | $<10^{-4}$ | $<10^{-4}$ |
| 10 Cycles, Training + Extinction | | $<10^{-4}$ | $<10^{-4}$ | $<10^{-4}$ | $<10^{-4}$ |
| *Figure 4B* | DANi1> CsChrimson | | | | |
| 20x Pre-Test/Post-Test | | $<10^{-4}$ | $<10^{-4}$ | $<10^{-4}$ | $<10^{-4}$ |
| 20x Pre-Test/Next Day | | $<10^{-4}$ | $<10^{-4}$ | $<10^{-4}$ | $<10^{-4}$ |
| 20x (Only Test Next Day) Pre-Test/Next Day | | $<10^{-4}$ | $<10^{-4}$ | $<10^{-4}$ | $<10^{-4}$ |
| 5x Pre-Test/Post-Test | | $<10^{-4}$ | $<10^{-4}$ | $<10^{-4}$ | $<10^{-4}$ |
| 5x Pre-Test/Next Day | | $<10^{-4}$ | $<10^{-4}$ | $<10^{-4}$ | $<10^{-4}$ |
| 2x Pre-Test/Post-Test | | $<10^{-4}$ | $<10^{-4}$ | $<10^{-4}$ | $<10^{-4}$ |
| 2x Pre-Test/Next Day | | 0.2086 | 0.0501 | 0.3524 | 0.07216 |
| 2x (Only Test Next Day) Pre-Test/Next Day | | $<10^{-4}$ | $<10^{-4}$ | $<10^{-4}$ | $<10^{-4}$ |
| No Train Pre-Test/Post-Test | | 0.4035 | 0.3319 | 0.7893 | 0.2003 |
| No Train Pre-Test/Next Day | | 0.1583 | 0.0530 | 0.3071 | 0.8884 |
| RP 20x Pre-Test/Post-Test | | 0.2677 | 0.1507 | 0.4276 | 0.7396 |
| RP 20x Pre-Test/Next Day | | 0.4205 | 0.3481 | 0.8474 | 0.3765 |
| Ext Post-Train Pre-Test/Next Day | | 0.1801 | 0.0146 | 0.3315 | 0.01336 |
| Ext Pre-Test Pre-Test/Next Day | | $<10^{-4}$ | $<10^{-4}$ | $<10^{-4}$ | $<10^{-4}$ |
| *Figure 4C* | DANi1> CsChrimson | | | | |
| M 20x (CXM+/ATR+) Pre-Test/Post-Test | | $<10^{-4}$ | $<10^{-4}$ | $<10^{-4}$ | $<10^{-4}$ |
| M 20x (CXM+/ATR+) Pre-Test/Next Day | | $<10^{-4}$ | $<10^{-4}$ | $<10^{-4}$ | $<10^{-4}$ |
| M 20x (CXM-/ATR+) Pre-Test/Post-Test | | $<10^{-4}$ | $<10^{-4}$ | $<10^{-4}$ | $<10^{-4}$ |
| M 20x (CXM-/ATR+) Pre-Test/Next Day | | $<10^{-4}$ | $<10^{-4}$ | $<10^{-4}$ | $<10^{-4}$ |
| S 10x (CXM+/ATR+) Pre-Test/Next Day | | 0.1099 | 0.014 | 0.1671 | 0.02985 |
| S 10x (CXM-/ATR+) Pre-Test/Next Day | | $<10^{-4}$ | $<10^{-4}$ | $<10^{-4}$ | $<10^{-4}$ |
| M 10x (CXM+/ATR+) Pre-Test/Next Day | | $<10^{-4}$ | $<10^{-4}$ | $<10^{-4}$ | $<10^{-4}$ |
| M 10x (CXM-/ATR+) Pre-Test/Next Day | | $<10^{-4}$ | $<10^{-4}$ | $<10^{-4}$ | $<10^{-4}$ |
| *Figure 4D* | DANi1> hs-dCREB2-b; CsChrimson | | | | |
| M 10x, HS Pre-Test/Post-Test | | $<10^{-4}$ | $<10^{-4}$ | $<10^{-4}$ | $<10^{-4}$ |
| M 10x, HS Pre-Test/Next Day | | $<10^{-4}$ | $<10^{-4}$ | $<10^{-4}$ | $<10^{-4}$ |
| M 10x, No HS Pre-Test/Post-Test | | $<10^{-4}$ | $<10^{-4}$ | $<10^{-4}$ | $<10^{-4}$ |
| M 10x, No HS Pre-Test/Next Day | | $<10^{-4}$ | $<10^{-4}$ | $<10^{-4}$ | $<10^{-4}$ |
| S 10x, HS Pre-Test/Post-Test | | 0.3804 | 0.2830 | 0.7310 | 0.2750 |
| S 10x, HS Pre-Test/Next Day | | 0.2645 | 0.08860 | 0.4650 | 0.3802 |
| S 10x, No HS Pre-Test/Post-Test | | $<10^{-4}$ | $<10^{-4}$ | $<10^{-4}$ | $<10^{-4}$ |
| S 10x, No HS Pre-Test/Next Day | | $<10^{-4}$ | $<10^{-4}$ | $<10^{-4}$ | $<10^{-4}$ |

larva chose the $CO_2$ containing channel was $p(n_c,j)$. Then we sought a set of parameters $\theta$ that maximized

$$\sum_{n_c}\sum_j \log P(p(n_c,j)|n(n_c,j),\theta) \tag{4}$$

where $P$ was the model specific-probability function. For instance, for the quantized learning (two-Gaussian shifting fraction) model:

$$P(p(n_c,j)|n(n_c,j),\theta) = f_u(n_c)\mathcal{N}\left(p(n_c,j),\mu_u,\tilde{\sigma}\sqrt{\frac{\mu_u*(1-\mu_u)}{n(n_c,j)}}\right) +$$
$$(1-f_u(n_c))\mathcal{N}\left(p(n_c,j),\mu_t,\tilde{\sigma}\sqrt{\frac{\mu_t*(1-\mu_t)}{n(n_c,j)}}\right) \tag{5}$$

where $\mathcal{N}(x,\mu,\sigma) = \frac{1}{\sqrt{2\pi\sigma^2}}\exp-\frac{(x-\mu)^2}{2\sigma^2}$ and the parameters $\theta$ are

$$\theta = \{\mu_u,\mu_t,\tilde{\sigma},f_u(0),f_u(1),f_u(2),f_u(3),f_u(4),f_u(5),f_u(10),f_u(20)\} \tag{6}$$

The parameter $\tilde{\sigma}$ represents an adjustment to the expected variance due to counting statistics. If all larva chose randomly and independently from the two channels with a fixed probability $\bar{p}$ of choosing $CO_2$, then we would expect that the number of times the $CO_2$ containing channel would be binomially distributed. For ease of computation, we approximated the binomial distribution as a normal distribution. In this case, the probability density of observing $p(n_c,j)$ given $n(n_c,j)$ would be normally distributed with mean $\bar{p}$ and variance

$$\sigma^2 = \frac{(\bar{p})(1-\bar{p})}{n(n_c,j)} \tag{7}$$

In fact, we found that after choosing a $CO_2$ containing channel, both naive and trained larvae are less likely to choose the $CO_2$ containing channel the next time they approach the intersection. Because the choices are not independent, the variance of the mean of a series of choices is not given by *Equation 7*. Instead, we modeled the variance as

$$\sigma^2 = \tilde{\sigma}^2\frac{(\bar{p})(1-\bar{p})}{n(n_c,j)} \tag{8}$$

where $\tilde{\sigma}$ was a global fit parameter in the shifting and exponential fraction models and in the shifting mean model a function of the amount training. This formulation preserves the properties that the variance should increase as the mean probability of choosing $CO_2$ approaches 50% and should be larger when fewer decisions are averaged together. However, if we instead just assume a single global σ, the results of our analysis (that the exponential fraction model is preferred) are unchanged.

In the graded learning (single Gaussian with shifting mean and variance) model, μ and σ were allowed to change as a function of training. The probability of an individual observation was

$$P(p(n_c,j)|n(n_c,j),\theta) = \mathcal{N}\left(p(n_c,j),\mu(n_c),\frac{\sigma(n_c)}{\sqrt{n(n_c,j)}}\right) \tag{9}$$

and the parameters were

$$\theta = \{\mu(0),\sigma(0),\mu(1),\sigma(1),\mu(2),\sigma(2),\mu(3),\sigma(3),\mu(4),\sigma(4),\mu(5),\sigma(5),\mu(10),\sigma(10),\mu(20),\sigma(20)\} \tag{10}$$

The exponential fraction model is identical to the quantized learning model, except that the fraction of untrained larvae is an exponentially decreasing function of the number of training cycles:

$$f_u(n_c) = \lambda^{n_c} \tag{11}$$

and the parameters were

$$\theta = \{\mu_u,\mu_t,\tilde{\sigma},\lambda\} \tag{12}$$

These models were then fit to the data by maximizing the log-likelihood of the observed data set using the MATLAB function fmincon. The predictions of these fits are shown in *Figure 2*. These results are summarized in *Table 4*, along with the Aikake and Bayes Information Criterion, AIC and BIC, which are used to compare models with different numbers of parameters. According to both AIC and BIC, the exponential fraction model is strongly favored.

Throughout the paper 'Fraction of larvae trained' represents the best fit to the two Gaussian shifting fraction model. The error bars represent the uncertainty in the model fit. Specifically, they represent the range of $f$ over which

**Table 4.** Model fits to data in *Figure 2*.

Shifting Mean and $\tilde{\sigma}$, shifting fraction, and exponential fraction models are presented in *Figure 2*. Model name: name of the model. Formula: expression for the probability of the data given the model and its parameters. # params: number of free parameters in the model. $\Delta \log(P)$ logarithm of the probability of the data given best fit to this model minus logarithm of the probability of the data given the best fit model overall. A higher (less negative) value means the model better fits the data without regard to the number of parameters. $\Delta AIC$, $\Delta BIC$ - Aikake and Bayes Information Criterion minus the lowest values over the models tested. Lower numbers indicate model is favored. According to both criterion, the exponential fraction model is strongly favored over the shifting fraction model, and the shifting fraction model is strongly favored over all models except the exponential fractional model.

| Model name | Formula | # params | $\Delta \log(P)$ | $\Delta AIC$ | $\Delta BIC$ |
|---|---|---|---|---|---|
| Shifting Mean (fixed $\tilde{\sigma}$) | $P \propto \prod\limits_{n_c \in (0,1,2,3,4,5,10,20)} \prod\limits_{j} \mathcal{N}\left( p(n_c,j), \mu(n_c), \tilde{\sigma}\sqrt{\dfrac{\mu(n_c)*(1-\mu(n_c))}{n(n_c,j)}} \right)$ | 9 | −42.7 | 84.86 | 104.45 |
| Shifting Mean and $\sigma$ (Graded learning) | $P \propto \prod\limits_{n_c \in (0,1,2,3,4,5,10,20)} \prod\limits_{j} \mathcal{N}\left( p(n_c,j), \mu(n_c), \dfrac{\sigma(n_c)}{\sqrt{n(n_c,j)}} \right)$ | 16 | −12.9 | 39.3 | 86.3 |
| Shifting Fraction (Quantized learning) | $P \propto \prod\limits_{n_c \in (0,1,2,3,4,5,10,20)} \prod\limits_{j} f_u(n_c)\mathcal{N}\left( p(n_c,j), \mu_u, \tilde{\sigma}\sqrt{\dfrac{\mu_u*(1-\mu_u)}{n(n_c,j)}} \right) + \\ \dots (1-f_u(n_c))\mathcal{N}\left( p(n_c,j), \mu_t, \tilde{\sigma}\sqrt{\dfrac{\mu_t*(1-\mu_t)}{n(n_c,j)}} \right)$ | 11 | −3.93 | 11.3 | 38.7 |
| Shifting Fraction (3 clusters) | $P \propto \prod\limits_{n_c \in (0,1,2,3,4,5,10,20)} \prod\limits_{j} f_1(n_c)\mathcal{N}\left( p(n_c,j), \mu_1, \tilde{\sigma}\sqrt{\dfrac{\mu_1*(1-\mu_1)}{n(n_c,j)}} \right) + \\ \dots f_2(n_c)\mathcal{N}\left( p(n_c,j), \mu_2, \tilde{\sigma}\sqrt{\dfrac{\mu_2*(1-\mu_2)}{n(n_c,j)}} \right) + \\ \dots (1-f_1(n_c)-f_2(n_c))\mathcal{N}\left( p(n_c,j), \mu_3, \tilde{\sigma}\sqrt{\dfrac{\mu_3*(1-\mu_3)}{n(n_c,j)}} \right)$ | 20 | 0 | 21.4 | 84.1 |
| Exponential Fraction (All-or-none) | $P \propto \prod\limits_{n_c \in (0,1,2,3,4,5,10,20)} \prod\limits_{j} \lambda^{n_c}\mathcal{N}\left( p(n_c,j), \mu_u, \tilde{\sigma}\sqrt{\dfrac{\mu_u*(1-\mu_u)}{n(n_c,j)}} \right) + \\ \dots (1-\lambda^{n_c})\mathcal{N}\left( p(n_c,j), \mu_t, \tilde{\sigma}\sqrt{\dfrac{\mu_t*(1-\mu_t)}{n(n_c,j)}} \right)$ | 4 | −5.3 | 0 | 0 |

| Symbol | Definition | Symbol | Definition |
|---|---|---|---|
| $n_c$ | number of training cycles | $p(n_c,j)$ | fraction of times $j^{th}$ larva chose $CO_2$ after $n_c$ cycles |
| $\mu(n_c)$ | mean probability of choosing $CO_2$ after $n_c$ training cycles | $n(n_c,j)$ | # choices made by $j^{th}$ larva after $n_c$ training cycles |
| $\tilde{\sigma}$ | global adjustment to binomial standard deviation | $\sigma(n_c)$ | training dependent standard deviation |
| $\mu_u$ | probability of larva in untrained group choosing $CO_2$ | $\mu_t$ | probability of larva in trained group choosing $CO_2$ |
| $f_u(n_c)$ | fraction of larvae in untrained group after $n_c$ cycles | $\mu_1, \mu_2, \mu_3$ | probability of larva in group 1,2,3 choosing $CO_2$ |
| $f_1(n_c), f_2(n_c)$ | fraction of larvae in groups 1,2 after $n_c$ cycles | $\lambda$ | fraction of larvae not trained after one cycle |
| $\mathcal{N}(x,\mu,\sigma)$ | normal cdf: $\dfrac{1}{\sqrt{2\pi\sigma^2}}e^{-\frac{(x-\mu)^2}{2\sigma^2}}$ | $\Delta \log(P)$ | relative log probability of data given model |
| AIC | Aikake Information Criterion: $2k - 2\log(P)$, k = # params | $\Delta AIC$ | AIC - lowest AIC |
| BIC | Bayes Information Criterion: $k \log(n_A) - 2\log(P)$, k = # params, $n_A$ = # animals | $\Delta BIC$ | BIC - lowest BIC |

$$\log P(\text{data}|\theta_0, f) \geq \log P(\text{data}|\theta_0, f_0) - \frac{1}{2} \tag{13}$$

where $f$ is the fraction of trained larvae, $f_0$, is the best fit fraction of trained larvae, and $\theta_0$ represents the best fit of the remainder of the parameters, which are not adjusted.

## Acknowledgements

We thank Jerry Yin for 17–2 hs-dCREB2-b and Marta Zlatic for SS00864. This project was supported by NSF grant 1455015, NIH grant DP2-EB022359, and a Sloan Foundation fellowship to MHG. The funders had no role in the design or analysis of the experiments. The following ORCIDs apply to the authors: 0000-0001-6611-5941 (AL), and 0000-0001-7528-6101 (MG).

## Additional information

### Funding

| Funder | Grant reference number | Author |
|---|---|---|
| National Institutes of Health | 1DP2EB022359 | Amanda Lesar<br>Javan Tahir<br>Jason Wolk<br>Marc Gershow |
| National Science Foundation | 1455015 | Amanda Lesar<br>Javan Tahir<br>Jason Wolk<br>Marc Gershow |
| Alfred P. Sloan Foundation | | Marc Gershow |

The funders had no role in study design, data collection and interpretation, or the decision to submit the work for publication.

### Author contributions

Amanda Lesar, Conceptualization, Data curation, Software, Formal analysis, Investigation, Methodology, Writing - original draft, Writing - review and editing; Javan Tahir, Software; Jason Wolk, Investigation; Marc Gershow, Conceptualization, Formal analysis, Supervision, Funding acquisition, Visualization, Methodology, Writing - original draft, Project administration, Writing - review and editing

### Author ORCIDs

Amanda Lesar (iD) https://orcid.org/0000-0001-6611-5941
Marc Gershow (iD) https://orcid.org/0000-0001-7528-6101

### Decision letter and Author response

Decision letter https://doi.org/10.7554/eLife.70317.sa1
Author response https://doi.org/10.7554/eLife.70317.sa2

## Additional files

### Supplementary files

- Supplementary file 1. Schematics and production files for y-maze components.
- Transparent reporting form

## Data availability

Summary statistics are included as a supplemental table in the article. Animal by animal choices in temporal sequence are provided as supplemental spreadsheets. Video files have been deposited in Dryad.

The following dataset was generated:

| Author(s) | Year | Dataset title | Dataset URL | Database and Identifier |
|---|---|---|---|---|
| Lesar A, Tahir J, Wolk J, Gershow M | 2021 | Switch-like and persistent learning in individual *Drosophila* larvae | https://doi.org/10.5061/dryad.hqbzkh1gs | Dryad Digital Repository, 10.5061/dryad.hqbzkh1gs |

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
