## [Decision Letter]

**Acceptance summary:**

Over the past two decades, the *Drosophila* larva has proven to be an advantageous system to study the neural basis of memory and its effects on orientation behavior. While larvae clearly learn, this behavior has been mostly characterized through en masse assays. To this date, it has been extremely difficult – if not impossible – to characterize learning at the level of single larvae. Here the authors present a tour-de-force assay, controlling the frequency and the exact timing of the presentation of the conditioned and unconditioned signals. With their new assay, they demonstrate the switch-like nature of learning in individual larvae, an important finding. Their work revisits multiple aspects of the theory of associative learning in the *Drosophila* larva, including the role of repeated training, the emergence of memory extinction, and the overnight consolidation of memory. This manuscript will have a major impact on the field of memory and learning in *Drosophila* and in the field more broadly.

**Decision letter after peer review:**

Thank you for submitting your article "Switch-like and persistent memory formation in individual *Drosophila* larvae" for consideration by *eLife*. Your article has been reviewed by 3 peer reviewers, and the evaluation has been overseen by a Reviewing Editor and Ronald Calabrese as the Senior Editor. The reviewers have opted to remain anonymous.

The reviewers have discussed their reviews with one another, and they were generally enthusiastic about the work's technical achievements and its connections to our understanding and memory. There were several areas where the work could improve, however, and the Reviewing Editor has drafted the following revisions to help you prepare a revised submission.

Essential revisions:

1. The choice of CO_2_ as a CS is both a curse and a blessing. The experimentalists must overcome innate avoidance of the signal, instead of the value of the signal being neutral to a naive animal. The authors speculate that the conditioning here is through inhibition of avoidance, and the picture they try to build (and it would be useful to have this as a simple mathematical model, rather than just a picture) is that an unconditioned optogenetic stimulus decreases avoidance of the conditioned stimulus. This is not the standard Pavlovian scheme, where, traditionally, positive reinforcement increases preferences (+/++) and negative reinforcement increases avoidance (-/+-) or decreases preference (-/-+). Instead, it's an unusual structure where positive reinforcement decreases avoidance (+/--). This is uncommon -- and results in precisely the same behavior limitations that the authors noted: the most one can do is to decrease avoidance to zero, and then the subsequent presentation of CS/US pairs does not lead to the emergence of the preference. The reviewers thought that the manuscript would become stronger if the authors tried to speculate what aspects of the animal's ecology would make this uncommon functional organization favored.

2. Potentially, a bigger issue is that the training in these experiments lasts for a very short time (from 30 s to 15 min or so), while the readout of the behavioral preference takes an hour, during which many unrewarded presentations of CS happen. In the paper, the authors themselves show that unrewarded CS presentations lead to a reduction in the behavioral response (Figure 3), to the point that overnight memory consolidation is not observed (Figure 4). Thus this long scale of the assay compared to the time scale of dynamics of the learning and extinction themselves makes interpretation of the findings very hard, at least for me. For example, is the 50% maximum choice of CO2 due to the animal not being able to establish the preference to it (and only being able to suppress the avoidance), or is it because the animal establishes a strong preference, which then gets partially washed away during the one hour of testing? There are a few ways that this and similar concerns can be addressed. First, a different assay can be established, where the preference is measured as quickly as it gets established and extinguished. Given *eLife*'s general prohibition on asking for additional experiments, however, one could instead explore if the preference of animals does not change during the course of the testing phase. This could be done by analyzing the preference over fifteen-minute segments and checking for drift (one could even combine animals to do so). Third, one can try to establish a mathematical model of conditioning and extinction, which would account for unrewarded CS presentations, and then see whether all of the data can be explained within this model. Or maybe one can do something totally different -- but I believe that some analysis of the effects of the assay on the conditioning state must be performed.

3. The authors talk about the quantized response as compared to gradual learning. This makes it seem that there are only two states that the animals can be in. But this is, in fact, unclear from the data. It's clear that there are two modes: indifferent to CO_2_ and avoiding it, but the modes are wide. Is there an additional signal there? Where is the width of the modes coming from? Is it simply the counting statistics of making, on average, pN out of N choices? Or are the data hiding something more interesting? This could be addressed by being a bit more careful with statistical analysis, and not treating the data as being fit by two Gaussians with arbitrary widths, but as a mixture of two Bernoulli distributions -- would such a model work? If not, then why?

---

## [Author Response]

Essential revisions:1. The choice of CO_2_ as a CS is both a curse and a blessing. The experimentalists must overcome innate avoidance of the signal, instead of the value of the signal being neutral to a naive animal.

We agree that the choice of CO_2_ as the CS, compared to the standard panel of odorants normally used, requires us to be careful in interpreting results and comparing them to the usual paradigm. While naive larvae are indifferent to but can be trained to approach linalool, the effect is much smaller than for more commonly used odors [Saumweber et al., 2011]. Most studies of larval learning, including important recent work demonstrating DAN-i1 activation as a reward [Saumweber et al., 2018, Thum and Gerber, 2019, Schleyer et al., 2020, Weiglein et al., 2019, Eschbach et al., 2020a] use innately attractive odors. While two attractive odors can be titrated and balanced against each other to get an initially neutral untrained behavior [Saumweber et al., 2011], the more common approach is to use a reciprocal paradigm which compares the preferences of oppositely trained groups, eliminating the need for a neutral baseline condition [Gerber and Stocker, 2006]. Therefore we would respectfully argue that the difference between using CO_2_ and the more common paradigms is not that we are using an odor with an innate valence, but that the innate valence is negative, rather than positive.

We have revised the text to make clear the difference in the innate valence:

“Activation of the DAN-i1 pair of mushroom body input neurons has been shown to act as a reward for associative learning [Saumweber et al., 2018, Thum and Gerber, 2019, Schleyer et al., 2020, Weiglein et al., 2019, Eschbach et al., 2020a]. In these experiments, the conditioned odor was innately attractive, but CO_2_ is innately aversive. We wondered whether pairing DAN-i1 activation with CO_2_ would lessen or even reverse the larva’s innate avoidance of CO_2_.”

The authors speculate that the conditioning here is through inhibition of avoidance, and the picture they try to build (and it would be useful to have this as a simple mathematical model, rather than just a picture) is that an unconditioned optogenetic stimulus decreases avoidance of the conditioned stimulus.

We use "decrease avoidance" in a strictly descriptive sense. Individual larvae initially avoid CO_2_ and following training no longer avoid it. At a population level, the population’s avoidance of CO_2_ is decreased with each successive presentation of CO_2_, but the population never shows an attraction to CO_2_. Given that we never observed a statistically significant attraction to CO_2_ on either the individual or population level, we felt that "decreased avoidance" was a more accurate description than "increased attraction." Of course if one defines avoidance to be the negative of attraction, then the two formulations are mathematically equivalent.

There is an emerging model of how the MB encodes and executes learned navigational behaviors. In this model, some MBONs encode approach and other avoidance. Appetitive training reduces the drive a CS provides to the avoidance promoting MBONs, resulting in approach. So in some sense, according to this model, all appetitive conditioning results from "inhibition of avoidance." In the discussion, we now place our work in the context of this model:

“While this work does not directly speak to the neural mechanism behind the change in preference, it is congruent with the evolving model of learning in *Drosophila*. […] Why in our experiments the learned appetitive drive appears to exactly cancel but not overcome the innate aversion should be the subject of further study; it may be a simple coincidence or artifact of the experimental protocol, or it may reflect more profound circuit principles.”

This is not the standard Pavlovian scheme, where, traditionally, positive reinforcement increases preferences (+/++) and negative reinforcement increases avoidance (-/+-) or decreases preference (-/-+). Instead, it's an unusual structure where positive reinforcement decreases avoidance (+/--). This is uncommon -- and results in precisely the same behavior limitations that the authors noted: the most one can do is to decrease avoidance to zero, and then the subsequent presentation of CS/US pairs does not lead to the emergence of the preference.

We are unsure how the fact that CO_2_ is innately aversive prevents larvae from developing a preference for CO_2_ following repeated training. In agreement with the point from Reviewer #1’s public review, we were surprised that repeated positive reinforcement did not lead to eventual attraction. If this could be clarified or a reference provided, we would be happy to address it in our discussion.

The reviewers thought that the manuscript would become stronger if the authors tried to speculate what aspects of the animal's ecology would make this uncommon functional organization favored.

If we had to speculate as to what aspects of the larva’s ecology lead to innate CO_2_ avoidance at all concentrations, we would guess that CO_2_ might signal the presence of a predator or of overcrowding and eventual lack of oxygen. For a burrowing animal, the latter is especially important, and avoiding CO_2_ might be a way to get out of a closed pocket before suffering respiratory distress. However, this is just speculation. Given that the model organism had many generations to adapt to a non-ecological setting, and that neither the training nor the testing has an ecological basis, we are hesitant to even guess whether, for instance, wild larvae do not learn to approach CO_2_ given appropriate reinforcement in a natural setting. If we were forced to speculate, one reason to learn to approach CO_2_ is if it signals the presence of food. Food should by itself have an innately attractive odor, so perhaps eliminating avoidance of CO_2_ is sufficient to increase the ability of the larva to locate food.

2. Potentially, a bigger issue is that the training in these experiments lasts for a very short time (from 30 s to 15 min or so), while the readout of the behavioral preference takes an hour, during which many unrewarded presentations of CS happen. In the paper, the authors themselves show that unrewarded CS presentations lead to a reduction in the behavioral response (Figure 3), to the point that overnight memory consolidation is not observed (Figure 4). Thus this long scale of the assay compared to the time scale of dynamics of the learning and extinction themselves makes interpretation of the findings very hard, at least for me. For example, is the 50% maximum choice of CO2 due to the animal not being able to establish the preference to it (and only being able to suppress the avoidance), or is it because the animal establishes a strong preference, which then gets partially washed away during the one hour of testing? There are a few ways that this and similar concerns can be addressed. First, a different assay can be established, where the preference is measured as quickly as it gets established and extinguished. Given eLife's general prohibition on asking for additional experiments, however, one could instead explore if the preference of animals does not change during the course of the testing phase. This could be done by analyzing the preference over fifteen-minute segments and checking for drift (one could even combine animals to do so). Third, one can try to establish a mathematical model of conditioning and extinction, which would account for unrewarded CS presentations, and then see whether all of the data can be explained within this model. Or maybe one can do something totally different -- but I believe that some analysis of the effects of the assay on the conditioning state must be performed.

As requested, we tested whether the larvae expressed a different preference immediately following training. Following 2,5, and 20 cycles of training, we quantified the population average response in the first 10 minutes following training and for the first 5 decisions, regardless of how quickly they were made. In neither case did we see evidence that the initial response differed from the long-time response, and in particular, we did not find evidence that trained larvae exhibited a preference for CO_2_ immediately following training. We also analyzed the post-training behavioral readout in 15 minute increments and did not see any clear temporal signature. Finally we carried out a new experiment in which we "refreshed" the training every 15 minutes to overcome any effects of extinction; we did not see increased attraction in this case either. This data appears as a supplement to figure 2 and is discussed in the text as follows:

“Given the relatively short duration of training and the ability of unrewarded CO_2_ presentations to extinguish prior training, we wondered whether larvae might change their CO_2_ preferences over the course of the hour-long post-training behavioral readout. […] Thus we concluded that the apparent limit of 50% population preference to CO_2_ following training was not due to the long time-scale of the behavioral readout.”

We understand there is some tension between the stability of the behavior during testing following two cycles of training and the fact that this testing period completely abolishes the formation of ARM. It is already clear in the literature that there is significant complexity in the different phases of memory formation, consolidation, and expression in both larval and adult *Drosophila*, and it is entirely plausible that the unrewarded presentations during the behavioral test do not affect immediate memory expression but do prevent consolidation to ARM. It is also plausible that some other experimental factor (e.g. earlier removal to food for overnight storage in the absence of behavioral test) might explain the results. Because of its ability to precisely control the timing and nature of unrewarded presentations, our apparatus will allow us to study precisely these questions in greater detail in the future.

We have added a paragraph addressing extinction to the discussion:

“We directly measured the ability of unrewarded CO_2_ presentations to extinguish a just-formed memory by presenting CO_2_ without air immediately following training. […] Further study will be required to confirm this. Our apparatus can precisely control the timing and nature of both rewarded and unrewarded presentations to probe different phases of memory formation and consolidation.”

3. The authors talk about the quantized response as compared to gradual learning. This makes it seem that there are only two states that the animals can be in. But this is, in fact, unclear from the data. It's clear that there are two modes: indifferent to CO2 and avoiding it, but the modes are wide. Is there an additional signal there? Where is the width of the modes coming from? Is it simply the counting statistics of making, on average, pN out of N choices? Or are the data hiding something more interesting? This could be addressed by being a bit more careful with statistical analysis, and not treating the data as being fit by two Gaussians with arbitrary widths, but as a mixture of two Bernoulli distributions -- would such a model work? If not, then why?

The width of the peaks is explained by counting statistics. They are actually somewhat narrower than one would expect from binomial statistics alone, because the decisions larvae make in the y-maze are not fully independent of each other (there is some tendency to choose an air channel following a choice of CO_2_).